PREPARED FOR SUBMISSION TO JHEP

HRI/ST/1702
HU-EP-17/07
HU-MATH-
2017/01
ICTS/2017/2

# Implications of the AdS/CFT Correspondence on Spacetime and Worldsheet OPE Coefficients

Sudip Ghosh,[a] Sourav Sarkar[b,c] and Mritunjay Verma[a,d,e]

[a]*International Centre for Theoretical Sciences, TIFR, Hesaraghatta, Hubli, Bengaluru-560089, India*

[b]*Institut für Mathematik und Institut für Physik, Humboldt-Universität zu Berlin, IRIS-Adlershof, Zum Großen Windkanal 6, 12489 Berlin, Germany*

[c]*Max-Planck-Institut für Gravitationsphysik, Albert-Einstein-Institut, Am Mühlenberg 1, 14476 Potsdam, Germany*

[d]*Harish-Chandra Research Institute, Chhatnag Road, Jhunsi, Allahabad-211019, India*

[e]*Homi Bhabha National Institute, Training School Complex, Anushakti Nagar, Mumbai 400085, India*

*E-mail:* sudip.ghosh@icts.res.in, sarkar@physik.hu-berlin.de, mritunjayverma@hri.res.in

ABSTRACT: We explore the connection between the operator product expansion (OPE) in the boundary and worldsheet conformal field theories in the context of $\text{AdS}_{d+1}/\text{CFT}_d$ correspondence. Considering single trace scalar operators in the boundary theory and using the saddle point analysis of the worldsheet OPE [1], we derive an explicit relation between OPE coefficients in the boundary and worldsheet theories for the contribution of single trace spin $\ell$ operators to the OPE. We also consider external vector operators and obtain the relation between OPE coefficients for the exchange of single trace scalar operators in the OPE. We revisit the relationship between the bulk cubic couplings in the Supergravity approximation and the OPE coefficients in the dual boundary theory. Our results match with the known examples from the case of $\text{AdS}_3/\text{CFT}_2$. For the operators whose two and three point correlators enjoy a non renormalization theorem, this gives a set of three way relations between the bulk cubic couplings in supergravity and the OPE coefficients in the boundary and worldsheet theories.

# 1 Introduction

The AdS/CFT correspondence gives a non-perturbative duality between string theories in AdS spacetimes and a certain class of quantum field theories living on the asymptotic boundary. Even though there is no explicit proof of the duality till now, there have been numerous checks during the last two decades which are consistent with the conjectured duality. However, these explicit checks have mostly been performed in the regime where the bulk spacetime is weakly curved and the dual string theory is well approximated by classical Supergravity. In order to systematically venture beyond this regime, it is certainly desirable to have a worldsheet formulation. But the worldsheet description is often not easily tractable. For example, the presence of RR fluxes complicates the quantization of the worldsheet sigma model in the usual RNS formalism. Consequently, this obstructs the direct construction of vertex operators.

The case of $AdS_3/CFT_2$, however, offers some examples where both the worldsheet theory and the boundary CFT theories are more explicitly understood. This has made it possible to perform several calculations that can check the duality beyond the usual Supergravity regime [2–14]. However, in other examples of the AdS/CFT correspondence, the worldsheet theory has been less tractable. A particularly interesting case is the class of free Yang Mills theories on the boundary and there have been various attempts and proposals to understand the worldsheet description of such theories [15–21]. However, in this case also, a complete control over the worldsheet sigma model is still lacking.

Fortunately, it turns out that just the assumption of the existence of a well defined worldsheet description can reproduce basic features of the dual field theory. In this paper, we shall focus on an important aspect of any local quantum field theory which is the existence of an operator product expansion (OPE). The OPE allows the product of nearby local operators to be expanded in a sum of local operators. For the CFTs, this expansion has nice convergence properties [22].

For the class of CFTs with weakly coupled AdS string duals, it was shown in [1] that the OPE can be understood from the perspective of the worldsheet theory. The analysis was based on very general properties of the connection between the boundary and worldsheet operators and did not rely on any specific theory. In particular, it was shown that the contribution of single trace scalar operators in the OPE of two boundary scalar operators can be reproduced from the OPE of the corresponding worldsheet vertex operators. In this paper, we perform a generalisation of this result to the case where operators with non-zero spin are exchanged in the OPE of boundary scalar operators and also consider the situation in which the external operators are conserved spin 1 currents in the boundary CFT. Making use of relation between the boundary and worldsheet correlators in the OPE regime, we write down an explicit relationship between the OPE coefficients in the boundary and the worldsheet theories in a model independent way.

In the Supergravity regime of the bulk String theory, the boundary correlation functions can be computed using Witten diagrams. The cubic coupling constants of the bulk effective field theory can be related to 3-point functions coefficients in the boundary by calculating

the associated three point Witten diagrams. Also, in the case of 4-point functions there has been significant work in analysing the OPE limit of the Witten diagrams and its matching with the corresponding expected behaviour in the boundary CFT [23–31]. For operators whose 2 and 3 point correlation functions do not receive quantum corrections as the coupling strength is varied, such as superconformal chiral primaries, our general analysis enables us to connect with these calculations and obtain a trio of relations among the boundary OPE coefficients, bulk couplings and the worldsheet OPE coefficients.

The rest of this paper is organised as follows. In section 2, we briefly review the saddle point analysis of [1]. The main result of this analysis is that the OPE of single trace operators in the boundary theory can be reproduced from a saddle point analysis in the worldsheet theory. In section 3, we consider the worldsheet vertex operators which are dual to boundary spinning operators and take into account their contribution to the worldsheet OPE of two scalar vertex operators. In this section, we also give a nice interpretation to the structure of the OPE expansion in terms of the so called shadow operators. In section 4, we consider a 4-point function of scalars in the boundary theory and using the results of sections 2 and 3, obtain an explicit relationship between OPE coefficients in the boundary and worldsheet CFTs for the contribution of traceless symmetric spin $\ell$ operators to the OPE. In section 5, we carry out a similar analysis for external conserved vector currents with a scalar exchange in their OPE. In section 6, we note some known results relating boundary OPE coefficients and bulk AdS couplings. Finally, in section 7, we shall compare our general results with some of the relevant results in the case of $AdS_3/CFT_2$ correspondence and end with conclusion and some future outlook in section 8. Appendices will elaborate on our notations and conventions and contain some of the calculations and some useful results needed in the analysis.

It is common in the $AdS_3/CFT_2$ literature to denote the scaling dimensions of the boundary and the worldsheet theories by $2j$ and $2\Delta$ respectively (see, e.g., [1, 7]). However, we shall use a convention which is more common in other situations, namely, our boundary and worldsheet scaling dimensions will be denoted by $\Delta$ and $h$ respectively.

## 2    Review of Saddle point Analysis on the World-Sheet

In this section we briefly review the general analysis of [1] which relates the operator product expansions in the boundary CFT and the worldsheet CFT describing the dual bulk string theory. We consider Euclidean CFTs in $d$-dimensions which can admit dual descriptions in terms of weakly coupled string theories on $AdS_{d+1} \times M$, where $M$ is a compact manifold. The vertex operators on the worldsheet which create perturbative single string states can be related to a special class of local operators in the dual boundary CFT by the AdS/CFT correspondence as[1]

$$\mathcal{O}_{\Delta,q}(x) = \int d^2z \; \mathcal{V}_{\Delta,q}(x; z, \bar{z}) \tag{2.1}$$

---

[1]We leave the relative normalization between worldsheet and boundary CFT operators implicit for now. We shall take this into account in section 4.

In the above expression, the integration is over the worldsheet co-ordinates. $\mathcal{V}$ denotes the worldsheet vertex operator, $\Delta$ is the scaling dimension of the boundary operator $\mathcal{O}$ and $q$ denotes quantum numbers for additional possible global symmetries of the boundary CFT. The worldsheet scaling dimension of $\mathcal{V}$ will be denoted by $h(\Delta, q)$[2]. Following [1], we shall use the terminology of large $N$ gauge theories to refer to the operators $\mathcal{O}_{\Delta,q}$ (which create single string states) as "single-trace" operators.

The vertex operators in the physical Hilbert space of the worldsheet CFT are labelled by $\Delta = \frac{d}{2} + 2is$, $s \in \mathbb{R}$. In the boundary CFT, these correspond to the principal series representation of the $d$-dimensional Euclidean conformal group $SO(d+1,1)$ and, by the standard AdS/CFT dictionary, are dual to normalizable modes in the bulk AdS spacetime[3]. The most general form of scalar contribution to the OPE of two such vertex operators can be written as

$$\mathcal{V}_{\Delta_1, q_1}(x, z)\mathcal{V}_{\Delta_2, q_2}(0, 0) = \sum_q \int_C d\Delta \int d^d x' F(z; x, x'; \Delta_i, \Delta; q_i, q)\mathcal{V}_{\Delta, q}(x'; 0) + \cdots \quad (2.2)$$

The label $C$ in this expression denotes that the $\Delta$ integral is along the contour $\frac{d}{2} + 2is$ and the dots denote the contribution from descendants of the worldsheet vertex operator $\mathcal{V}_{\Delta,q}$. In order to avoid cluttering the notation, we shall henceforth denote the external vertex operators simply as $\mathcal{V}_1$ and $\mathcal{V}_2$.

We now want to relate this to the OPE in the dual boundary CFT. However we are interested in boundary operators belonging to unitary representations of $SO(d+1,1)$ which are labelled by real values of $\Delta$ satisfying the unitarity bound $\Delta \geq \frac{d}{2}$. These operators are dual to non-normalizable modes in AdS and consequently are not present in the physical Hilbert space of states in the worldsheet CFT. Thus, we need to analytically continue the above OPE expression to real values of $\Delta_1, \Delta_2$ and $\Delta$. Generally, as a result of such analytic continuation, the function $F(z; x, x'; \Delta_i, \Delta; q_i, q)$ can develop poles and this will yield additional contributions to the OPE in (2.2). In some cases, these can be shown to be related to the contribution of "multi-trace" operators to the boundary OPEs [7]. In this work, we shall not concern ourselves with the analysis of such "multi-trace" contributions. We proceed with the assumption that apart from these extra contributions, the above form of the worldsheet OPE is preserved.

We also note that the operator $\mathcal{V}_{\Delta,q}$ appearing in the OPE (2.2) has the interpretation of a scalar operator in the boundary CFT. In general, the worldsheet OPE will receive contributions from vertex operators which can carry boundary co-ordinate indices. Such operators have the natural interpretation in the boundary theory as operators with spin. We shall consider these in the next section.

Now using the conformal symmetries of the worldsheet and the boundary theories,

---

[2]Here $h$ is the sum of the left and right worldsheet conformal dimensions. As mentioned in introduction, our conventions differ from those used in [1]. See appendix A for more details.

[3]In an interesting recent paper [32], operators belonging to principal series representations in Euclidean CFTs were considered for studying general solutions to the conformal bootstrap equations for arbitrary Lie Groups.

we can fix the form of the function $F(z; x, x'; \Delta_i, \Delta; q_i, q)$ appearing in (2.2) upto a factor which will be related to the 2-point and 3-point function coefficients of the worldsheet CFT. Ignoring the contributions of the worldsheet descendant operators, we have

$$\mathcal{V}_1(x, z)\mathcal{V}_2(0, 0) = \sum_q \int_C d\Delta \int d^d x' \frac{|z|^{-(h(1)+h(2)-h(\Delta,q))}}{|x|^\alpha |x'|^\beta |x' - x|^\gamma} F(\Delta_i, \Delta; q_i, q)\mathcal{V}_{\Delta,q}(x'; 0) \quad (2.3)$$

The parameters $\alpha, \beta, \gamma$ can be determined by demanding invariance of the above OPE under boundary conformal transformations as shown in the next section. In particular invariance under scale transformations in the boundary CFT implies

$$\alpha + \beta + \gamma - d = \Delta_1 + \Delta_2 - \Delta \quad (2.4)$$

Also, the vertex operators which create physical string excitations must be level matched Virasoro primaries with worldsheet conformal dimensions $(1, 1)$. Thus,

$$h(1) + h(2) = 4 \quad (2.5)$$

In what follows, we shall be interested in analysing the small $|x|$ limit of the above OPE since this corresponds to the OPE limit in the boundary CFT. Changing to new coordinate $y$ such that $x' = y|x|$ and keeping only the leading order terms in this limit, we obtain

$$\mathcal{V}_1(x, z)\mathcal{V}_2(0, 0) = \sum_q \int_C d\Delta \frac{|z|^{-(h(1)+h(2)-h(\Delta,q))}}{|x|^{\alpha+\beta+\gamma-d}} F_{12\Delta}^{(0,0,0)} \, \mathcal{V}_{\Delta,q}(0; 0) \int \frac{d^d y}{|y|^\beta |y - \hat{x}|^\gamma} \quad (2.6)$$

where, following the convention introduced in appendix A, we have changed the notation as $F_{12\Delta}^{(0,0,0)} \equiv F(\Delta_i, \Delta; q_i, q)$. The three zeros in the superscript denote the boundary spacetime spins of the three operators which appear in the above OPE equation.

Consider now the $n$-point correlation functions of the boundary CFT operators. By the AdS/CFT duality, these can be expressed as correlation functions of the integrated worldsheet vertex operators as,

$$\mathcal{A}_n \equiv \left\langle \mathcal{O}_1(x_1)\mathcal{O}_2(0) \prod_{i=3}^{n-2} \mathcal{O}_i(x_i)\mathcal{O}_{n-1}(x_{n-1})\mathcal{O}_n(x_n) \right\rangle$$

$$= \int d^2 z \left\langle \mathcal{V}_1(x_1; z)\bar{c}c\mathcal{V}_2(0; 0) \prod_{i=3}^{n-2} \int d^2 w_i \mathcal{V}_i(x_i; w_i)\bar{c}c\mathcal{V}_{n-1}(x_{n-1}; 1)\bar{c}c\mathcal{V}_n(x_n; \infty) \right\rangle_{S^2} \quad (2.7)$$

where, using the global part of the worldsheet conformal symmetry group, we have set three of the vertex operator insertions on the worldsheet at $0, 1$ and $\infty$. The $c$ and $\bar{c}$ are the usual ghost and anti-ghost insertions which are required to make unintegrated vertex operators BRST invariant. The label $S^2$ in the worldsheet correlator above implies that we shall consider tree level or genus 0 contributions to the worldsheet correlation functions. In other words, this analysis captures the planar contribution (i.e. large $N$-limit of the boundary CFT) to the boundary correlation function.

Using the worldsheet OPE (2.6) and the relations (2.4), (2.5), we obtain on performing the angular worldsheet integration

$$\mathcal{A}_n = |x_1|^{-\Delta_1 - \Delta_2} \sum_q \int d\ln|z| \int_C d\Delta |z|^{h(\Delta,q)-2} |x_1|^\Delta B(\Delta_i, q_i; \Delta, q) \quad (2.8)$$

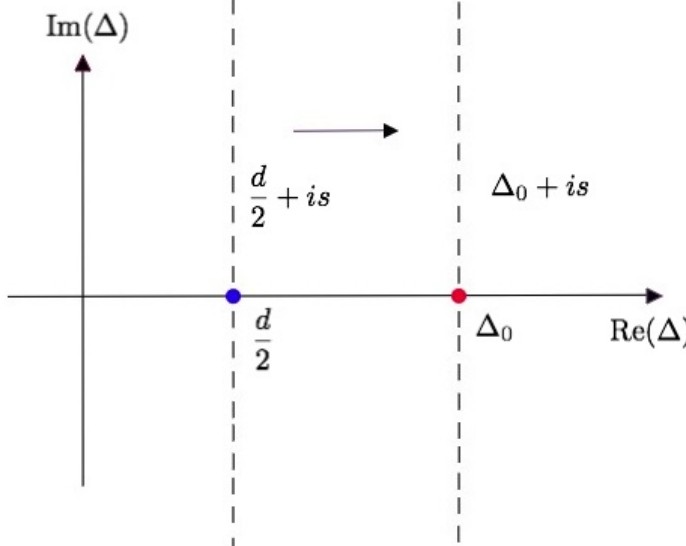

**Figure 1**. (Borrowed from [1]) The vertical dashed line labelled by $\frac{d}{2} + is$ denotes the contour of integration in the OPE of normalizable vertex operators. The original contour is deformed along the real axis as indicated by the arrow such that it intersects the point $\Delta = \Delta_0$ (marked in red), which is the saddle point of the $\Delta$ integral in equation (2.8).

where,

$$B(\Delta_i, q_i; \Delta, q) \equiv 2\pi F_{12\Delta}^{(0,0,0)} \langle \bar{c}c\mathcal{V}_{\Delta,q}(0;0)X \rangle \int \frac{d^d y}{|y|^\beta |y - \hat{x_1}|^\gamma} \qquad (2.9)$$

$X$ in the above expression denotes all the other $n-3$ integrated vertex operators and the two fixed vertex operators of equation (2.7).

Although not exact, the expression (2.8) yields the most dominant contribution in the region of small $|z|$ and small $|x_1|$. The $\Delta$-integral in (2.8) can be shown to admit a saddle point at $\Delta = \Delta_0$ such that

$$\partial_\Delta h(\Delta, q)|_{\Delta=\Delta_0} = -2\,\frac{\ln|x_1|}{\ln|z|}$$

It turns out that the values of $\Delta_0$ satisfying this condition, in general, do not lie on the original contour of integration. Assuming the integrand to be an analytic function of $\Delta$, except at the location of specific poles, we can deform the integration contour such that it intersects the real $\Delta$-axis at $\Delta_0$ as illustrated in figure 1. As discussed in [1], the poles which we may cross while shifting the contour, don't change the conclusion of this analysis.

Having performed the $\Delta$ integral via saddle point, we can again resort to a saddle point approximation for evaluating the $z$-integral. The saddle point condition for the $z$ integral gives

$$h(\Delta_0, q) = 2$$

This is precisely the condition which must hold for a worldsheet vertex operator to correspond to a physical string state in the bulk AdS target space. Consequently this corresponds to a single trace boundary operator via the AdS/CFT duality.

To ensure that the saddle approximation is justified, we also need to check that the higher order fluctuations around the saddle value are suppressed. This indeed turns out to be the case provided $\partial_\Delta h|_{\Delta_0} < 0$ and $\partial_\Delta^2 h|_{\Delta_0} < 0$. A straightforward computation of the gaussian fluctuations around the saddle point finally gives the most dominant contribution to the integral in the spacetime OPE limit $|x_1| \to 0$ as,

$$\mathcal{A}_n = -2i\pi \sum_q \frac{|x_1|^{\Delta_0 - \Delta_1 - \Delta_2}}{\partial_\Delta h(\Delta, q)|_{\Delta = \Delta_0}} B(\Delta_i, q_i; \Delta_0, q) \tag{2.10}$$

The dependence of the above expression on $|x_1|$ and the $n-1$ point correlator in the boundary CFT through the $n-1$ point worldsheet correlator in $B(\Delta_i, q_i; \Delta_0, q)$, is exactly the form we expect when we consider the contribution to the boundary $n$-point function from the exchange of a single trace operator of dimension $\Delta_0$ in the OPE of $\mathcal{O}_{\Delta_1}$ and $\mathcal{O}_{\Delta_2}$.

Upto now, we have considered the contribution of only scalar operators in the OPE. In section 4, we shall generalise this to include the contribution of operators with spin. The saddle point analysis remains exactly identical to that described above. Hence, we shall not present the details of the saddle point analysis any further. Instead, our focus will be on extracting the relation between the OPE coefficients in the worldsheet and boundary theories by expressing (2.10) for the special case of 4-point functions in a factorized form in the (12)(34) channel. However, before that, in the next section, we consider the contributions of the more general operators in the OPE of two worldsheet (and also boundary) scalars.

## 3  General Scalar-Scalar Worldsheet OPE

In this section, we generalize the scalar-scalar OPE (2.2) to include the contributions of the worldsheet operators which are dual to boundary CFT operators with spin. As we shall see, the conformal invariance fixes the co-ordinate dependence of the OPE completely. We start by showing that the OPE structure can be given a nice interpretation in terms of shadow operators.

### 3.1  Structure of the OPE and Shadow Operators

We start by making a connection between the shadow operators and the structure of the OPE (2.2). We had noted in the previous section that the $z$ dependence of the function $F(z; x, x'; \Delta_i, \Delta; q_i, q)$ in (2.2) is fixed by worldsheet conformal invariance to be $|z|^{-(h(1)+h(2)-h(\Delta,q))}$. Thus, we can write (working with arbitrary values of $x_i$ and $z_i$ for the moment)

$$\mathcal{V}_1(x_1, z_1)\mathcal{V}_2(x_2, z_2) = \sum_q \int_C d\Delta \int d^d x |z_{12}|^{-(h(1)+h(2)-h(\Delta,q))} F_{12\Delta}^{(0,0,0)} L(x_1, x_2; x) \mathcal{V}_{\Delta,q}(x, z_2)$$

$$\tag{3.1}$$

The function $L(x_1, x_2; x)$ in the above equation denotes the functional dependence of $F(z_i; x_i, x; \Delta_i, \Delta; q_i, q)$ on $x_1, x_2$ and $x$. It should be noted that $L(x_1, x_2; x)$ is an entirely kinematic factor which can be determined from the symmetries of the theory. In particular we shall now show that the shadow operator formalism provides an efficient way of determining $L(x_1, x_2; x)$.

In a $d$-dimensional CFT, the shadow of a conformal primary operator $\mathcal{O}_\Delta^{\mu_1 \cdots \mu_\ell}$ of spin $\ell$ and scaling dimension $\Delta$ is defined as [33]

$$\tilde{\mathcal{O}}_{d-\Delta}^{\mu_1 \cdots \mu_\ell}(x) \equiv \frac{k_{\Delta,\ell}}{\pi^{d/2}} \int d^d y \, \frac{1}{|x-y|^{2(d-\Delta)}} J_{(\nu_1}^{\mu_1} \cdots J_{\nu_\ell)}^{\mu_\ell}(x-y) \mathcal{O}_\Delta^{\nu_1 \cdots \nu_\ell}(y) \tag{3.2}$$

where,

$$J^{\mu\nu}(x) = \delta^{\mu\nu} - 2\frac{x^\mu x^\nu}{x^2}$$

and the brackets in the subscript of $J_{(\nu_1}^{\mu_1} \cdots J_{\nu_\ell)}^{\mu_\ell}$ denote total symmetrization in those indices. The normalisation factor

$$k_{\Delta,\ell} \equiv \frac{1}{(\Delta-1)_\ell} \frac{\Gamma(d-\Delta+\ell)}{\Gamma(\frac{2\Delta-d}{2})}$$

ensures that

$$\tilde{\tilde{\mathcal{O}}}_{\Delta, \mu_1 \cdots \mu_\ell}^{(\ell)} = \mathcal{O}_{\Delta, \mu_1 \cdots \mu_\ell}^{(\ell)}$$

Now if we perform a conformal transformation in the boundary CFT, a scalar primary operator will transform as

$$\mathcal{O}_{\Delta,q}(x) \to [\Omega(x)]^\Delta \, \mathcal{O}_{\Delta,q}(x)$$

where $\Omega(x)$ is the conformal factor. This implies that under the boundary conformal transformation, the dual worldsheet vertex operator $\mathcal{V}_{\Delta,q}$ transforms as

$$\mathcal{V}_{\Delta,q}(z,x) \to [\Omega(x)]^\Delta \, \mathcal{V}_{\Delta,q}(z,x) \tag{3.3}$$

Since the boundary conformal symmetry acts as a global symmetry on the worldsheet, (3.1) should transform in a covariant fashion when a conformal transformation is implemented in the boundary CFT. Hence, using (3.3) we must have

$$L(x_1, x_2; x) \to [\Omega(x_1)]^{\Delta_1} [\Omega(x_2)]^{\Delta_2} [\Omega(x)]^{d-\Delta} L(x_1, x_2; x)$$

But this is precisely the conformal transformation property of a 3-point function involving the boundary operators $\mathcal{O}_{\Delta_1}, \mathcal{O}_{\Delta_2}$ and the shadow operator $\tilde{\mathcal{O}}_{d-\Delta}$. Thus

$$L(x_1, x_2; x) \propto \left\langle \mathcal{O}_{\Delta_1}(x_1) \mathcal{O}_{\Delta_2}(x_2) \tilde{\mathcal{O}}_{d-\Delta}(x) \right\rangle \tag{3.4}$$

Using (B.2) and (G.2), this 3-point function can be computed as [33]

$$
\begin{aligned}
&\langle \mathcal{O}_{\Delta_1}(x_1)\mathcal{O}_{\Delta_2}(x_2)\tilde{\mathcal{O}}_{d-\Delta}(x)\rangle \\
&= \frac{k_{\Delta,0}}{\pi^{d/2}}\int d^d y \, \frac{1}{|x-y|^{2(d-\Delta)}}\langle \mathcal{O}_{\Delta_1}(x_1)\mathcal{O}_{\Delta_2}(x_2)\mathcal{O}_{\Delta}(y)\rangle \\
&= \frac{k_{\Delta,0}}{\pi^{d/2}}\int d^d y \, \frac{1}{|x-y|^{2(d-\Delta)}}\left[\frac{\bar{C}^{(0,0,0)}_{\Delta_1\Delta_2\Delta}}{|y-x_1|^{\Delta+\Delta_1-\Delta_2}|y-x_2|^{\Delta+\Delta_2-\Delta_1}|x_1-x_2|^{\Delta_1+\Delta_2-\Delta}}\right] \\
&= \frac{H(\Delta_1,\Delta_2,\Delta,d)}{|x_1-x_2|^{\Delta_1+\Delta_2+\Delta-d}|x_2-x|^{\Delta_2-\Delta_1-\Delta+d}|x-x_1|^{\Delta_1-\Delta_2-\Delta+d}}
\end{aligned}
$$

where, $\bar{C}^{(0,0,0)}_{\Delta_1\Delta_2\Delta}$ is the three point function coefficient appearing in $\langle \mathcal{O}_{\Delta_1}(x_1)\mathcal{O}_{\Delta_2}(x_2)\mathcal{O}_{\Delta}(y)\rangle$ and

$$
H(\Delta_1,\Delta_2,\Delta,d) \equiv k_{\Delta,0}\bar{C}^{(0,0,0)}_{\Delta_1\Delta_2\Delta}\frac{\Gamma\left(\frac{\Delta_1-\Delta_2-\Delta+d}{2}\right)\Gamma\left(\frac{\Delta_2-\Delta_1-\Delta+d}{2}\right)\Gamma\left(\frac{2\Delta-d}{2}\right)}{\Gamma\left(\frac{\Delta+\Delta_1-\Delta_2}{2}\right)\Gamma\left(\frac{\Delta+\Delta_2-\Delta_1}{2}\right)\Gamma(d-\Delta)}
$$

Now, since we are only interested in obtaining the dependence of $L(x_1,x_2;x)$ on $x_1, x_2$ and $x$, we choose $(H(\Delta_1,\Delta_2,\Delta,d))^{-1}$ to be the proportionality constant in (3.4). Note that this is justified since equation (3.4) is a purely kinematical relation which follows from the fact that the boundary conformal symmetry manifests itself as a global symmetry of the worldsheet theory. In other words there is no dynamical information about the worldsheet theory in (3.4). Thus, we get

$$
\mathcal{V}_1(x_1,z_1)\mathcal{V}_2(x_2,z_2) = \sum_q \int_C d\Delta \int d^d x' \, \frac{|z_{12}|^{-(h(1)+h(2)-h(\Delta,q))}}{|x_{12}|^\alpha |x_2-x|^\beta |x-x_1|^\gamma}F^{(0,0,0)}_{12\Delta}\,\mathcal{V}_{\Delta,q}(x;z_2)
$$

which is the same as (2.3) if we put $z_1=z, z_2=0, x_1=x, x_2=0$ and $x=x'$ above. Moreover, the values of $\alpha, \beta$ and $\gamma$ is fixed to be

$$
\begin{aligned}
\alpha &= \Delta_1 + \Delta_2 + \Delta - d,\\
\beta &= \Delta_2 - \Delta_1 - \Delta + d,\\
\gamma &= \Delta_1 - \Delta_2 - \Delta + d
\end{aligned}
$$

In appendix F, we shall obtain the same results for $\alpha, \beta$ and $\gamma$ in a slightly different way.

## 3.2 General World-Sheet OPE

We now consider the contributions of the operators of the form $\mathcal{V}^{\mu_1\cdots\mu_\ell}_{\Delta,q}(x,z)$ to the OPE between the two worldsheet scalar vertex operators $\mathcal{V}_1(x_1;z_1)$ and $\mathcal{V}_2(x_2;z_2)$ . The additional coordinate indices $(\mu_1,\cdots\mu_\ell)$ imply that the corresponding dual operator in the boundary CFT has spin $\ell$. From the worldsheet perspective, these can be interpreted as appropriate global symmetry labels. The most general form of the worldsheet OPE involving the

exchange of such operators can be written as

$$\mathcal{V}_1(x_1, z_1)\mathcal{V}_2(x_2, z_2)$$
$$= \sum_q \int_C d\Delta \int d^d x \, F_{\mu_1 \cdots \mu_\ell}(x_i, x; z_i; \Delta_i, \Delta; q_i, q)\mathcal{V}_{\Delta,q}^{\mu_1 \cdots \mu_\ell}(x, z_1)$$
$$= \sum_q \int_C d\Delta \int d^d x \, |z_{12}|^{-(h(1)+h(2)-h(\Delta,q))} F_{12\Delta}^{(0,0,\ell)} L_{\mu_1 \cdots \mu_\ell}(x_1, x_2, x)\mathcal{V}_{\Delta,q}^{\mu_1 \cdots \mu_\ell}(x, z_1) \quad (3.5)$$

In going to the last line, we have used the conformal symmetry of the worldsheet theory to fix the $z$ dependence. The function $L_{\mu_1 \cdots \mu_\ell}(x_1, x_2, x)$ is just a kinematic factor which can be determined completely using the symmetries of the boundary theory. In order to obtain the functional form of $L_{\mu_1 \cdots \mu_\ell}(x_1, x_2, x)$ we shall again make use of the shadow formalism. Consider applying an inversion on the boundary coordinates. This gives,

$$x_i^\mu \to \frac{x_i^\mu}{x_i^2} \quad , \quad x_{ij}^2 \to \frac{x_{ij}^2}{x_i^2 x_j^2} \quad , \quad d^d x \to \frac{d^d x}{|x|^{2d}}$$

A boundary CFT operator $\mathcal{O}_{\Delta,q}^{\mu_1 \cdots \mu_\ell}$ with spin $\ell$ and dimension $\Delta$ transforms under inversion as

$$\mathcal{O}_{\Delta,q}^{\mu_1 \cdots \mu_\ell}(x) \to |x|^{2\Delta} J_{\nu_1}^{\mu_1}(x) \cdots J_{\nu_\ell}^{\mu_\ell}(x) \, \mathcal{O}_{\Delta,q}^{\nu_1 \cdots \nu_\ell}(x)$$

The dual worldsheet operator $\mathcal{V}_{\Delta,q}^{\mu_1 \cdots \mu_\ell}(x, z)$ then tranforms according to

$$\mathcal{V}_{\Delta,q}^{\mu_1 \cdots \mu_\ell}(x, z) \to |x|^{2\Delta} J_{\nu_1}^{\mu_1}(x) \cdots J_{\nu_\ell}^{\mu_\ell}(x) \, \mathcal{V}_{\Delta,q}^{\nu_1 \cdots \nu_\ell}(x, z)$$

For $\ell = 0$ we will have

$$\mathcal{O}_{\Delta,q}(x) \to |x|^{2\Delta}\mathcal{O}_{\Delta,q}(x), \quad \mathcal{V}_{\Delta,q}(x, z) \to |x|^{2\Delta}\mathcal{V}_{\Delta,q}(x, z)$$

Using these relations it is evident that to preserve the form of the OPE in (3.5) under inversion, $L_{\mu_1 \cdots \mu_\ell}(x_1, x_2, x)$ must transform as

$$L_{\mu_1 \cdots \mu_\ell}(x_1, x_2, x) \to |x_1|^{2\Delta_1}|x_2|^{2\Delta_2}|x|^{2(d-\Delta)} \left( J_{\nu_1}^{\mu_1}(x) \cdots J_{\nu_\ell}^{\mu_\ell}(x) \right)^{-1} L_{\nu_1 \cdots \nu_\ell}(x_1, x_2, x)$$

where, the inverses of $J^{\mu\nu}s$ in the above expression can be obtained using the identity $J^{\mu\rho}(x)J_{\rho\nu}(x) = \delta^\mu_{\ \nu}$.

Now it is easy to verify that a 3 point function involving $\mathcal{O}_{\Delta_1}, \mathcal{O}_{\Delta_2}$ and $\tilde{\mathcal{O}}_{d-\Delta}^{\mu_1 \cdots \mu_\ell}$ transforms exactly in the above fashion under inversion which will imply,

$$L_{\mu_1 \cdots \mu_\ell}(x_1, x_2, x) \propto \left\langle \mathcal{O}_{\Delta_1}(x_1)\mathcal{O}_{\Delta_2}(x_2)\tilde{\mathcal{O}}_{d-\Delta}^{\mu_1 \cdots \mu_\ell}(x) \right\rangle \quad (3.6)$$

This 3-point function can be obtained using the result for $\left\langle \mathcal{O}_{\Delta_1}(x_1)\mathcal{O}_{\Delta_2}(x_2)\mathcal{O}_{d-\Delta}^{\mu_1 \cdots \mu_\ell}(x) \right\rangle$ given in equation (B.2), the definition of shadow operator (3.2) and the integral (G.3) as

[33]

$$\langle \mathcal{O}_{\Delta_1}(x_1)\mathcal{O}_{\Delta_2}(x_2)\tilde{\mathcal{O}}^{\mu_1\cdots\mu_\ell}_{d-\Delta}(x)\rangle$$

$$= \frac{k_{\Delta,\ell}}{\pi^{d/2}} \int d^d y \, \frac{J^{\mu_1}_{(\nu_1}\cdots J^{\mu_\ell}_{\nu_\ell)}(x-y)}{|x-y|^{2(d-\Delta)}} \langle \mathcal{O}_{\Delta_1}(x_1)\mathcal{O}_{\Delta_2}(x_2)\mathcal{O}^{\nu_1\cdots\nu_\ell}_{\Delta}(y)\rangle$$

$$= \frac{k_{\Delta,\ell}}{\pi^{d/2}} \int d^d y \, \frac{J^{\mu_1}_{(\nu_1}\cdots J^{\mu_\ell}_{\nu_\ell)}}{|x-y|^{2(d-\Delta)}} \left[ \frac{\bar{C}^{(0,0,\ell)}_{\Delta_1\Delta_2\Delta} Z^{\nu_1}(x_1,x_2,y)\cdots Z^{\nu_\ell}(x_1,x_2,y)}{|y-x_1|^{\Delta+\Delta_1-\Delta_2-\ell}|y-x_2|^{\Delta+\Delta_2-\Delta_1-\ell}|x_1-x_2|^{\Delta_1+\Delta_2-\Delta+\ell}} \right]$$

$$= \frac{H'(\Delta_1,\Delta_2,\Delta,d,\ell)\;\; Z^{\mu_1}(x_1,x_2,x)\cdots Z^{\mu_\ell}(x_1,x_2,x)}{|x_1-x_2|^{\Delta_1+\Delta_2+\Delta-d+\ell}|x_2-x|^{\Delta_2-\Delta_1-\Delta+d-\ell}|x-x_1|^{\Delta_1-\Delta_2-\Delta+d-\ell}}$$

where,

$$H'(\Delta_1,\Delta_2,\Delta,d,\ell) \equiv k_{\Delta,\ell}\bar{C}^{(0,0,\ell)}_{\Delta_1\Delta_2\Delta} \frac{\lambda_\ell \Gamma\left(\frac{\Delta_1-\Delta_2-\Delta+d+\ell}{2}\right)\Gamma\left(\frac{\Delta_2-\Delta_1-\Delta+d+\ell}{2}\right)\Gamma\left(\frac{2\Delta-d}{2}\right)}{\Gamma\left(\frac{\Delta+\Delta_1-\Delta_2+\ell}{2}\right)\Gamma\left(\frac{\Delta+\Delta_2-\Delta_1+\ell}{2}\right)\Gamma(d-\Delta+\ell)}$$

and,

$$Z^\mu(x_1,x_2,x_3) \equiv \frac{(x_1-x_3)^\mu}{(x_1-x_3)^2} - \frac{(x_2-x_3)^\mu}{(x_2-x_3)^2} \quad , \qquad \lambda_\ell(a,b) \equiv \prod_{r=0}^{\ell-1}\left(\frac{a+b}{2}+\ell-1+r\right)$$

The functions $Z^\mu$ and $\lambda_\ell$ will apper throughtout the draft with different arguments. We shall specify only the arguments from now on. In the expression of $H'$ above, we have

$$\lambda_\ell(a,b) = \lambda_\ell(\Delta+\Delta_1-\Delta_2-\ell, \Delta+\Delta_2-\Delta_1-\ell)$$

Once again, the object of interest is just the functional dependence of $L_{\mu_1\cdots\mu_\ell}(x_1,x_2,x)$ on its arguments. Thus, we take the proportionality factor in (3.6) to be $(H'(\Delta_1,\Delta_2,\Delta,d,\ell))^{-1}$. The worldsheet OPE for the exchange of operators dual to boundary spin $\ell$ operators can then be expressed as

$$\mathcal{V}_1(x_1,z_1)\mathcal{V}_2(x_2,z_2)$$
$$= \sum_q \int_C d\Delta \int d^d x \, \frac{|z_{12}|^{-(h_1+h_2-h_{\Delta,q})}\; Z_{\mu_1}\cdots Z_{\mu_\ell}}{|x_1-x_2|^\alpha |x_2-x|^\beta |x-x_1|^\gamma} F^{(0,0,\ell)}_{12\Delta} \mathcal{V}^{\mu_1\cdots\mu_\ell}_{\Delta,q}(x,z_1) \qquad (3.7)$$

where $\alpha,\beta,\gamma$ in the above expression are given by

$$\alpha = \Delta_1+\Delta_2+\Delta-d+\ell,$$
$$\beta = \Delta_2-\Delta_1-\Delta+d-\ell,$$
$$\gamma = \Delta_1-\Delta_2-\Delta+d-\ell \qquad (3.8)$$

Thus, we see that the dependence of the worldsheet OPE on the coordinates of the boundary spacetime can be rather efficiently obtained by appealing to the notion of shadow operators. In section 4, we shall show that the tensor structure in the worldsheet OPE obtained via

this method indeed reproduces the structure which is expected to appear in the OPE of the dual operators in the boundary CFT theory.

We should mention that one could have fixed the functional form of $L_{\mu_1\cdots\mu_\ell}$ in equation (3.5) by noting that the conformal invariance dictates the $x$ dependence of correlator involving two scalars and one spin $\ell$ object to be as given in equation (B.2). This would still leave the power of factors in the denominator of (B.2) undetermined which could then be fixed by demanding the OPE equation to be consistent with the conformal invariance. As shown in appendix F, this brute force way of computing $\alpha, \beta$ and $\gamma$ matches with the result given in (3.8).

We can simplify the expression in (3.7) by putting the vertex operator $\mathcal{V}_2$ at the origin of the $x$ and $z$ coordinate systems and change the integration variable to $y$ such that $x = y|x_1|$. This gives,

$$
\begin{aligned}
&\mathcal{V}_1(x_1, z_1)\mathcal{V}_2(0,0) \\
&= \sum_q \int_C d\Delta \frac{|z_1|^{-(h_1+h_2-h_{\Delta,q})}}{|x_1|^{\alpha+\beta+\gamma+\ell-d}} F_{12\Delta}^{(0,0,\ell)} \int d^d y \, \frac{Z_{\mu_1}(\hat{x}_1, 0, y)\cdots Z_{\mu_\ell}(\hat{x}_1, 0, y)}{|y|^\beta \, |y - \hat{x}_1|^\gamma} \mathcal{V}_{\Delta,q}^{\mu_1\cdots\mu_\ell}(y|x_1|, 0)
\end{aligned}
$$

Taylor expanding the exchanged operators around $|x_1| = 0$ and keeping only the leading terms (since we are ignoring the contributions of descendants anyway), we obtain

$$
\mathcal{V}_1(x_1, z_1)\mathcal{V}_2(0,0) = \sum_q \int_C d\Delta \frac{|z_1|^{-(h_1+h_2-h_{\Delta,q})}}{|x_1|^{\alpha+\beta+\gamma+\ell-d}} F_{12\Delta}^{(0,0,\ell)} \mathcal{V}_{\Delta,q}^{\mu_1\cdots\mu_\ell}(0,0) G_{\mu_1\cdots\mu_\ell}(x_1) \quad (3.9)
$$

where, using the integral (G.4), we have

$$
\begin{aligned}
G^{\mu_1\cdots\mu_\ell}(x_1) &\equiv \int d^d y \, \frac{Z_{\mu_1}(\hat{x}_1, 0, y)\cdots Z_{\mu_\ell}(\hat{x}_1, 0, y)}{|y|^\beta \, |y - \hat{x}_1|^\gamma} \\
&= \pi^{d/2} \lambda_\ell(\beta, \gamma) \frac{\Gamma\left(\frac{d-\beta}{2}\right)\Gamma\left(\frac{d-\gamma}{2}\right)\Gamma\left(\frac{\beta+\gamma-d}{2}+\ell\right)}{\Gamma\left(\frac{\beta}{2}+\ell\right)\Gamma\left(\frac{\gamma}{2}+\ell\right)\Gamma\left(\frac{2d-\beta-\gamma}{2}\right)} \left[\hat{x}_1^{\mu_1}\cdots\hat{x}^{\mu_\ell}\right] \quad (3.10)
\end{aligned}
$$

# 4 Relation between OPE coefficients in the Worldsheet and the Boundary theories

Suppose $C_{12\Delta}^{(0,0,\ell)}$ be the OPE coefficient which appears in the symmetric traceless spin $\ell$ contribution to the OPE of two boundary scalar operators. Our goal now is to obtain the relationship between $C_{12\Delta}^{(0,0,\ell)}$ and the worldsheet OPE coefficient $F_{12\Delta}^{(0,0,\ell)}$. To do this, we shall make use of the saddle point analysis reviewed in section 2.

We consider the 4-point correlator of the scalar operators of the boundary CFT. We can evaluate this correlator directly in the boundary theory in the OPE limit. Alternatively, we can use the representation of the boundary operators as integrated worldsheet vertex operators and use the saddle point analysis to evaluate it using the worldsheet theory. As

mentioned in section 2, the saddle point analysis gives the result in the OPE limit. Comparing the boundary and the worldsheet calculations will give us the desired relationship.

Since we intend to deal with relations between OPE coefficients across two theories, we need to fix the relative normalization between the vertex operators in the worldsheet theory and the operators in the boundary CFT. We can do this by comparing the two point function in the two theories. Choosing the coefficient of boundary two point function automatically fixes the normalization of the worldsheet vertex operators. This is done in Appendix E. Below, we shall work with the normalized vertex operators.

## 4.1 Boundary CFT Calculation

The four point function of arbitrary scalar operators in the boundary theory can be easily evaluated in the OPE limit. Using the OPE (C.1), the 3-point function (B.2) and the relation between the OPE coefficient and the 3-point function coefficient (D.1), we obtain

$$
\begin{aligned}
&\left\langle \mathcal{O}_1(x_1)\mathcal{O}_2(0)\mathcal{O}_3(x_3)\mathcal{O}_4(x_4)\right\rangle \\
&= \sum_{\Delta,q} C_{12\Delta}^{(0,0,\ell)} \frac{(x_1)_{\mu_1}\cdots(x_1)_{\mu_\ell}}{|x_1|^{(\Delta_1+\Delta_2-\Delta)+\ell}} \left\langle \mathcal{O}_{\Delta,q}^{\mu_1\cdots\mu_\ell}(0)\mathcal{O}_3(x_3)\mathcal{O}_4(x_4)\right\rangle \\
&= \sum_{\Delta,q} \frac{(\hat{x}_1)_{\mu_1}\cdots(\hat{x}_1)_{\mu_\ell}}{|x_1|^{\Delta_1+\Delta_2-\Delta}} \frac{K_\ell(\Delta)C_{12\Delta}^{(0,0,\ell)}C_{34\Delta}^{(0,0,\ell)}\left(Z^{\mu_1}(x_3,x_4,0)\cdots Z^{\mu_\ell}(x_3,x_4,0)-\text{Traces}\right)}{|x_3|^{\Delta_3+\Delta-\Delta_4-\ell}|x_4|^{\Delta_4+\Delta-\Delta_3-\ell}|x_{34}|^{\Delta_3+\Delta_4-\Delta+\ell}}
\end{aligned}
$$

$$(4.1)$$

where $K_\ell(\Delta)$ denotes the 2-point function coefficient of two spin $\ell$ operators as introduced in equation (B.1)[4] and $q$ denotes the quantum numbers due to additional global symmetries.

## 4.2 Worldsheet Calculation

Next, we calculate the same 4-point function in the OPE limit using the worldsheet theory. Unlike section 2, we shall now consider the contribution of symmetric traceless spin $\ell$ operators in the OPE. Following the procedure of section 2, the 4 point function is given in the OPE limit $(x_1 \to x_2 = 0)$ as follows

$$
\begin{aligned}
&\left\langle \mathcal{O}_1(x_1)\mathcal{O}_2(0)\mathcal{O}_3(x_3)\mathcal{O}_4(x_4)\right\rangle \\
&= \frac{1}{\prod_{i=1}^4 a_i^{(0)}} \int d^2z \langle \mathcal{V}_1(x_1;z)\bar{c}c\mathcal{V}_2(x_2;0)\bar{c}c\mathcal{V}_3(x_3;1)\bar{c}c\mathcal{V}_4(x_4;\infty)\rangle \\
&= -\frac{2i\pi}{\prod_{i=1}^4 a_i^{(0)}} \sum_q \frac{|x_1|^{\Delta_0-\Delta_1-\Delta_2}}{\partial_\Delta h(\Delta,q)|_{\Delta=\Delta_0}} \bar{B}(\Delta_i,q_i;\Delta_0,q)
\end{aligned}
$$

$$(4.2)$$

Here we have introduced the factors $a_i^{(0)}$ taking into account the relative normalization between $\mathcal{O}_i$ and $\mathcal{V}_i$. Their precise form is given in the appendix E. $\bar{B}$ is given by an

---

[4]Throughout the paper, we have chosen the 2-point function coefficient, namely $K_0(\Delta)$ to be equal to 1. This means that we shall not need to distinguish between the OPE coefficients $C_{123}^{(0,0,0)}$ and the corresponding three point function coefficient $\bar{C}_{123}^{(0,0,0)}$.

expression similar to the one given in equation (2.9) but now generalized to include the contribution from tensor operators

$$\bar{B} \equiv 2\pi F_{12\Delta_0}^{(0,0,\ell)} \left\langle \bar{c}c\mathcal{V}_{\Delta_0,q}^{\mu_1\cdots\mu_\ell}(0;0)\bar{c}c\mathcal{V}_3(x_3;1)\bar{c}c\mathcal{V}_4(x_4;\infty)\right\rangle G_{\mu_1\cdots\mu_\ell}(x_1) \tag{4.3}$$

$G^{\mu_1\cdots\mu_\ell}$ is given in equation (3.10) and the OPE coefficient $F_{12\Delta_0}^{(0,0,\ell)}$ is related to the two and three point function coefficients[5] on the worldsheet as derived in appendix D.2.

The 3-point correlator appearing in (4.3) can be written as the product of a ghost correlator and a matter correlator. The universal expression for matter correlator is fixed by the conformal invariance of the worldsheet and boundary theories and is given in equation (D.3). The ghost correlator is given by

$$\langle \bar{c}c(z_1)\bar{c}c(z_3)\bar{c}c(z_4)\rangle = C_{S_2}^g |z_{13}|^2|z_{34}|^2|z_{14}|^2 \tag{4.4}$$

where $C_{S_2}^g$ is the ghost normalization factor.

Fixing the points $z_1, z_3$ and $z_4$ in the ghost correlator (4.4) at $0, 1$ and $\infty$ respectively, using (D.3) for the matter correlator and the relation between the 3-point function coefficient and OPE coefficient given in equation (D.4), we obtain

$$\left\langle \bar{c}c\mathcal{V}_{\Delta_0,q}^{\mu_1\cdots\mu_\ell}(0;0)\bar{c}c\mathcal{V}_3(x_3;1)\bar{c}c\mathcal{V}_4(x_4;\infty)\right\rangle$$

$$= C_{S_2}^g \bar{F}_{34\Delta_0}^{(0,0,\ell)} \left[\lim_{z_4\to\infty} |z_4|^{-2(h_4-2)}\right] \frac{Z^{\mu_1}(x_3,x_4,0)\cdots Z^{\mu_\ell}(x_3,x_4,0) - \text{Traces}}{|x_3|^{(\Delta_3+\Delta_0-\Delta_4)-\ell}|x_4|^{(\Delta_4+\Delta_0-\Delta_3)-\ell}|x_3-x_4|^{(\Delta_3+\Delta_4-\Delta_0)+\ell}}$$

$$= F_{34\Delta_0}^{(0,0,\ell)} C_{S_2}^g \left[\pi^{d/2}\lambda_\ell(\beta',\gamma')D_\ell(\Delta_0)\frac{\Gamma\left(\frac{d-\beta'}{2}\right)\Gamma\left(\frac{d-\gamma'}{2}\right)\Gamma\left(\frac{\beta'+\gamma'-d}{2}+\ell\right)}{\Gamma\left(\frac{\beta'}{2}+\ell\right)\Gamma\left(\frac{\gamma'}{2}+\ell\right)\Gamma\left(\frac{2d-\beta'-\gamma'}{2}\right)}\right]$$

$$\times \frac{Z^{\mu_1}(x_3,x_4,0)\cdots Z^{\mu_\ell}(x_3,x_4,0) - \text{Traces}}{|x_3|^{(\Delta_3+\Delta_0-\Delta_4)-\ell}|x_4|^{(\Delta_4+\Delta_0-\Delta_3)-\ell}|x_3-x_4|^{(\Delta_3+\Delta_4-\Delta_0)+\ell}} \tag{4.5}$$

where $D_\ell(\Delta_0)$ is the 2-point function coefficient of vertex operators (see equation (D.2)) and,

$$\beta' \equiv \Delta_4 - \Delta_3 - \Delta_0 + d - \ell, \quad \gamma' \equiv \Delta_3 - \Delta_4 - \Delta_0 + d - \ell$$

Using the expression (4.5) in equation (4.3) and comparing equations (4.1) and (4.2), we finally obtain the following explicit relationship between the boundary OPE coefficients $C_{12\Delta_0}^{(0,0,\ell)}$ and worldsheet OPE coefficients $F_{12\Delta_0}^{(0,0,\ell)}$

$$\boxed{C_{12\Delta_0}^{(0,0,\ell)} = \left(\sqrt{\frac{-4i\pi^{d+2}C_{S_2}^g D_\ell(\Delta_0)}{\partial_\Delta h(\Delta,q)|_{\Delta=\Delta_0}K_\ell(\Delta_0)}}\, \frac{I_{12}}{a_1^{(0)}a_2^{(0)}}\right) F_{12\Delta_0}^{(0,0,\ell)}} \tag{4.6}$$

The above equation is the main result of our paper for the case of symmetric spin $\ell$ exchange. We remind the reader that in the above expression $C_{S_2}^g$ is the normalization of the ghost

---

[5]It is important to distinguish between the OPE coefficient $F_{12\Delta_0}^{(0,0,\ell)}$ and the 3-point function coefficient $\bar{F}_{12\Delta_0}^{(0,0,\ell)}$ since they are related by a non trivial factor. The relation between them is given in equation (D.4)

correlator defined through (4.4), $D_\ell(\Delta_0)$ is the coefficient in the 2-point function of the worldsheet operators $\mathcal{V}_{\Delta_0,q}$. $K_\ell(\Delta_0)$ appears as the 2-point function coefficient of boundary spin $\ell$ operators as defined in (B.1). The normalization factors $a_i^{(0)}$ are specified in (E.2) and $I_{12}$ is given by

$$I_{12} \equiv \frac{\lambda_\ell\, \Gamma\left(\frac{\Delta_0+\Delta_1-\Delta_2+\ell}{2}\right)\Gamma\left(\frac{\Delta_0+\Delta_2-\Delta_1+\ell}{2}\right)\Gamma\left(\frac{d-2\Delta_0}{2}\right)}{\Gamma\left(\frac{\Delta_1-\Delta_2-\Delta_0+d+\ell}{2}\right)\Gamma\left(\frac{\Delta_2-\Delta_1-\Delta_0+d+\ell}{2}\right)\Gamma\left(\ell+\Delta_0\right)} \tag{4.7}$$

The relation between the boundary OPE coefficients $C_{34\Delta_0}^{(0,0,\ell)}$ and the worldsheet OPE coefficients $F_{34\Delta_0}^{(0,0,\ell)}$ can be obtained by replacing (12) with (34) in (4.6), namely

$$\boxed{C_{34\Delta_0}^{(0,0,\ell)} = \left(\sqrt{\frac{-4i\pi^{d+2}C_{S_2}^g D_\ell(\Delta_0)}{\partial_\Delta h(\Delta,q)|_{\Delta=\Delta_0}K_\ell(\Delta_0)}}\,\frac{I_{34}}{a_3^{(0)}a_4^{(0)}}\right) F_{34\Delta_0}^{(0,0,\ell)}} \tag{4.8}$$

where, $I_{34}$ is given by an expression similar to (4.7) with the replacement of (12) by (34).

## 5 Generalization to External Spinning Operators

We can repeat the analysis of the previous section for correlation functions of spinning operators. For simplicity, we consider the 4-point function of two conserved vector currents and two scalar operators and consider the exchange of a scalar in the OPE of the two vectors. Our goal will again be to obtain the relationship between the OPE coefficients in the worldsheet and the boundary theories.

### 5.1 Boundary CFT Calculation

The conformal dimension of the conserved vector operators is $d-1$. Using the OPE (C.2), the 4-point function of two such operators and two scalar operators with scaling dimensions $\Delta_3$ and $\Delta_4$ can be evaluated as

$$\begin{aligned}
&\langle \mathcal{O}^\mu(x_1)\mathcal{O}^\nu(0)\mathcal{O}_3(x_3)\mathcal{O}_4(x_4)\rangle\\
&= \sum_\Delta C_{12\Delta}^{(1,1,0)}\frac{b\,\delta^{\mu\nu}+\hat{x}_1^\mu\hat{x}_1^\nu}{|x_1|^{2(d-1)-\Delta}}\langle\mathcal{O}_\Delta(0)\mathcal{O}_3(x_3)\mathcal{O}_4(x_4)\rangle\\
&= \sum_\Delta \frac{b\,\delta^{\mu\nu}+\hat{x}_1^\mu\hat{x}_1^\nu}{|x_1|^{2(d-1)-\Delta}}\frac{C_{12\Delta}^{(1,1,0)}C_{34\Delta}^{(0,0,0)}}{|x_3|^{\Delta_3+\Delta-\Delta_4}|x_4|^{\Delta_4+\Delta-\Delta_3}|x_{34}|^{\Delta_3+\Delta_4-\Delta}}
\end{aligned} \tag{5.1}$$

The constant $b$ is fixed for conserved currents and is given in equation (C.5).

### 5.2 Worldsheet Calculation

Again, we can represent the boundary operators as the integrated worldsheet vertex operators and calculate the desired 4-point function in the OPE limit using the saddle point

approximation. Using the normalization of the vertex operators (E.1) and the OPE (C.7), the saddle point analysis gives

$$
\begin{aligned}
&\left\langle \mathcal{O}^\mu(x_1)\mathcal{O}^\nu(0)\mathcal{O}_3(x_3)\mathcal{O}_4(x_4)\right\rangle \\
&= \frac{1}{a_1^{(1)}a_2^{(1)}a_3^{(0)}a_4^{(0)}} \int d^2z \langle \mathcal{V}^\mu(x_1;z)\bar{c}c\mathcal{V}^\nu(0;0)\bar{c}c\mathcal{V}_3(x_3;1)\bar{c}c\mathcal{V}_4(x_4;\infty)\rangle \\
&= -\frac{2i\pi}{a_1^{(1)}a_2^{(1)}a_3^{(0)}a_4^{(0)}} \sum_q \frac{|x_1|^{\Delta-2(d-1)}}{\partial_\Delta h(\Delta,q)|_{\Delta=\Delta_0}} B^{\mu\nu}(\Delta_i,q_i;\Delta_0,q)
\end{aligned}
\tag{5.2}
$$

where, the function $B^{\mu\nu}$ is given by

$$
B^{\mu\nu} \equiv 2\pi F_{12\Delta_0}^{(1,1,0)} \left\langle \bar{c}c\mathcal{V}_{\Delta,q}(0;0)\bar{c}c\mathcal{V}_3(x_3;1)\bar{c}c\mathcal{V}_4(x_4;\infty)\right\rangle H^{\mu\nu}(x_1)
\tag{5.3}
$$

$H^{\mu\nu}(x)$ is defined in equation (C.8).

The 3-point function appearing in (5.3) is a special case ($\ell = 0$) of (4.5). Hence, using (4.5) for $\ell = 0$, the expression of $H^{\mu\nu}$ given in (C.8) and comparing (5.2) with the corresponding boundary CFT result (5.1), we obtain the following relationships between the boundary and the worldsheet OPE coefficients

$$
\begin{aligned}
C_{12\Delta_0}^{(1,1,0)} &= \left( \sqrt{\frac{-4i\pi^{d+2}C_{S_2}^g D_0(\Delta_0)}{\partial_\Delta h(\Delta,q)|_{\Delta=\Delta_0}}} \; \frac{\bar{I}_{12}}{a_1^{(1)}a_2^{(1)}} \right) F_{12\Delta_0}^{(1,1,0)} \\
C_{34\Delta_0}^{(0,0,0)} &= \left( \sqrt{\frac{-4i\pi^{d+2}C_{S_2}^g D_0(\Delta_0)}{\partial_\Delta h(\Delta,q)|_{\Delta=\Delta_0}}} \; \frac{I_{34}}{a_3^{(0)}a_4^{(0)}} \right) F_{34\Delta_0}^{(0,0,0)}
\end{aligned}
\tag{5.4}
$$

where,

$$
\bar{I}_{12} \equiv \left[ \frac{2\Delta_0(2d-\Delta_0-2)}{(d-\Delta_0)^2(\Delta_0+1-d)} \right] \frac{\Gamma\left(\frac{\Delta_0}{2}\right)\Gamma\left(\frac{\Delta_0}{2}\right)\Gamma\left(\frac{d-2\Delta_0}{2}\right)}{\Gamma\left(\frac{d-\Delta_0}{2}\right)\Gamma\left(\frac{d-\Delta_0}{2}\right)\Gamma(\Delta_0)}
$$

and,

$$
I_{34} = \frac{\Gamma\left(\frac{\Delta_0+\Delta_3-\Delta_4}{2}\right)\Gamma\left(\frac{\Delta_0+\Delta_4-\Delta_3}{2}\right)\Gamma\left(\frac{d-2\Delta_0}{2}\right)}{\Gamma\left(\frac{\Delta_3-\Delta_4-\Delta_0+d}{2}\right)\Gamma\left(\frac{\Delta_4-\Delta_3-\Delta_0+d}{2}\right)\Gamma(\Delta_0)}
$$

Note that the result (5.4) is consistent with (4.8) for $\ell = 0$.

## 6   Relationship with AdS Coupling Constants

In this section, we quote some relations between cubic couplings in AdS supergravity and the OPE coefficients in the boundary CFT. These relations, for the coupling between two scalars $\phi_1$, $\phi_2$ and one spin $\ell$ operator $h^{\mu_1\cdots\mu_\ell}$, were obtained in [31]. The interaction term

is taken to be of the form $\frac{1}{S_{\phi\phi h}} g_{\phi\phi h}^{(0,0,\ell)} \phi_1 \left(\nabla_{\mu_1\cdots\mu_\ell}\phi_2\right) h^{\mu_1\cdots\mu_\ell}$, $S_{\phi\phi h}$ being a symmetry factor. We quote the result here,

$$C_{123}^{(0,0,\ell)}$$
$$= g_{\phi\phi h}^{(0,0,\ell)} \pi^{\frac{d}{2}} \sqrt{C_{\Delta_1} C_{\Delta_2} C_{\Delta_3,\ell}} \left[ \frac{\Gamma\left(\alpha_{12} + \frac{\ell}{2}\right)\Gamma\left(\alpha_{23} + \frac{\ell}{2}\right)\Gamma\left(\alpha_{13} + \frac{\ell}{2}\right)\Gamma\left(\frac{\Delta_1+\Delta_2+\Delta_3+\ell-d}{2}\right)}{2^{1-\ell}\Gamma(\Delta_1)\Gamma(\Delta_2)\Gamma(\Delta_3+\ell)} \right]$$
$$(6.1)$$

where, $\Delta_1, \Delta_2$ and $\Delta_3$ are the scaling dimensions of $\phi_1, \phi_2$ and $h^{\mu_1\cdots\mu_\ell}$ respectively and

$$\alpha_{12} \equiv \frac{\Delta_1 + \Delta_2 - \Delta_3}{2} \quad ; \quad \alpha_{23} \equiv \frac{\Delta_2 + \Delta_3 - \Delta_1}{2} \quad ; \quad \alpha_{13} \equiv \frac{\Delta_1 + \Delta_3 - \Delta_2}{2}$$

$C_{123}^{(0,0,\ell)}$ is the relevant OPE coefficient in the boundary CFT and $g_{\phi\phi h}^{(0,0,\ell)}$ is the cubic coupling constant in supergravity. $C_{\Delta,\ell}$ are the normalization factors associated with the bulk to boundary propagators $E_{\Delta,\ell}^{V_1 V_2 \cdots V_\ell}$ of spin $\ell$ operator given by

$$E_{\Delta,\ell}^{V_1 V_2 \cdots V_\ell}(P, X) = C_{\Delta,\ell} \frac{R^{V_1 V_2 \cdots V_\ell}}{(-2P \cdot X)^\Delta} \quad ; \quad C_{\Delta,\ell} = \frac{1}{2}\left(\frac{\ell + \Delta - 1}{\Delta - 1}\right)\frac{\pi^{-d/2}\Gamma(\Delta)}{\Gamma\left(\Delta + 1 - \frac{d}{2}\right)}$$

We have used the embedding space notation to express the propagator (see [31]). $R^{V_1 V_2 \cdots V_\ell}$ is the relevant tensor structure (dimensionless).

We shall also state the result for a Witten diagram with two external conserved currents (of dimension $d-1$)and a scalar. The interaction term (see [23]) considered is of the form $\eta^{\mu\nu}\eta^{\rho\sigma}\phi\partial_{[\mu}A_{\nu]}\partial_{[\rho}A_{\sigma]}$. This diagram can be computed by an inversion on the coordinates (which is an isometry of $AdS$) followed by direct evaluation of the integrals. The result between the corresponding OPE coefficient (see (B.3), (C.4)) and the cubic coupling in supergravity is given by,

$$C_{12\Delta}^{(1,1,0)} = g_{123}^{(1,1,0)} \pi^{\frac{d}{2}} \sqrt{C_{d-1,1} C_{d-1,1} C_\Delta} \frac{(d-2)^2 \Gamma\left(\frac{\Delta}{2} + 1\right)\Gamma\left(\frac{\Delta+d-2}{2}\right)\Gamma\left(\frac{2d-2-\Delta}{2}\right)}{2\Gamma(\Delta)\Gamma(d)\Gamma(d)}$$

The results presented here all assume a canonical normalization for terms in the Lagrangian as mentioned before. A factor of $\frac{1}{\sqrt{\prod_{i=1}^3 C_{\Delta_i,\ell_i}}}$ has been multiplied to each result from the Witten diagrams to ensure consistency with a unit normalized two point function. In general, we can have cubic vertices different from the ones considered here in which case the relation between the bulk cubic coupling and the OPE coefficient gets modified. Such results involving one or more higher spin gauge fields in the bulk can be obtained from work presented in [34–39].

## 7   Comparison with the case of AdS$_3$/CFT$_2$

One can use the AdS/CFT correspondence to compute the boundary CFT's correlation functions using the dual gravitational description. The computations in the boundary

theory are usually done in weak coupling regime whereas the calculations done in the bulk in the supergravity approximation give results in the strong coupling regime of the boundary CFT and there is, a priori, no reason to believe that the correlators computed at different coupling strengths should match with each other. However, it may happen that some correlators of a theory are independent of the coupling strength due to the supersymmetry. Matching of correlators in such cases is one of the important checks of the AdS/CFT correspondence.

In the example of duality between $\mathcal{N} = 4$ $SYM$ in four dimensions and string theory on $AdS_5 \times S^5$, the 3-point functions of chiral primaries calculated using the boundary theory and the bulk supergravity theory were shown to agree, thus indicating the existence of a non-renormalization theorem protecting the correlators [40] (see also [25–29, 41–46]). In this case, the worldsheet theory however is not well understood. Another interesting example is the correspondence between the S-dual description of IIB string theory living on $D1$-$D5$ with near horizon geometry $AdS_3 \times S^3 \times \mathcal{M}^4$ and the two dimensional $\mathcal{N} = (4, 4)$ CFT with deformed symmetric orbifold target space. It turns out that all the three corners of the picture, namely the two dimensional boundary conformal field theory, the bulk supergravity approximation and the worldsheet description of the 10 dimensional string theory, are tractable in this case (although, at different points in the moduli space). 3-point functions of chiral primaries have been shown to match from calculations done at all the three corners [8–10, 13]. This is not surprising now that a non-renormalization theorem for the 3-point functions of chiral primaries in two dimensional $\mathcal{N} = (4, 4)$ theory has been proved in [14, 47].

In this work, we have obtained relations between the OPE coefficients in the worldsheet theory and the dual boundary CFT and, in the previous section, noted the relations between the cubic couplings in the supergravity theory and the OPE coefficients in the boundary CFT dual to it. These relations are obtained in a model independent manner based on how we expect the vertex operators in the worldsheet theory and primaries in the boundary CFT to be related to each other [7]. For the operators whose 3-point functions in the boundary CFT are subject to a non-renormalization theorem, this triangle of relations closes. We shall now check that our results are consistent with some explicitly known results in the case of $AdS_3/CFT_2$.

We start by showing that our result for relation between the OPE coefficients of the boundary and the worldsheet theory match with the corresponding results for chiral primary operators of the $AdS_3/CFT_2$ case mentioned above. In [8], the three point function of the boundary CFT chiral primaries was computed via the three point function of the "dual" worldsheet vertex operators. This was shown to be in agreement with the results computed directly in the boundary CFT. To see that our calculations are consistent with the results for the chiral primaries of $AdS_3/CFT_2$, we shall check that the relation between 3-point function coefficients of the normalized operators on the boundary and the worldsheet theory matches with results in [8].

Using (4.6) and (D.4) (for $\ell = 0$), the relation between the 3-point function coefficients

in the boundary and the worldsheet theory in our computation, denoted $\bar{C}^{(0,0,0)}_{12\Delta_0}$ and $\bar{F}^{(0,0,0)}_{12\Delta_0}$ (for scalars) respectively, when applied to the case of $\text{AdS}_3/\text{CFT}_2$, is given by

$$\bar{C}^{(0,0,0)}_{12\Delta_0} = \left[ \frac{\sqrt{2\pi C^g_{S_2}}}{\sqrt{f(\Delta_1)D_0(\Delta_1)}\sqrt{f(\Delta_2)D_0(\Delta_2)}\sqrt{f(\Delta_0)D_0(\Delta_0)}} \right] \bar{F}^{(0,0,0)}_{12\Delta_0} \qquad (7.1)$$

where, we have used the fact that the factor $\frac{-2i\pi}{\partial_\Delta h(\Delta,q)|_{\Delta=\Delta_0}}$ appearing in (4.6) is just $\frac{1}{f(\Delta)}$, $f(\Delta)$ being $\Delta + 1$ in this case.

We now need to check that the computation performed in [8] for the 3-point function of chiral primaries matches with the above result. In [8], the chiral primaries considered are the $n$-cycle twist operators $\mathcal{O}_{\Delta,m,\bar{m}}$ of the boundary theory, where $\Delta = n-1$ is the operator dimension and $(m, \bar{m})$ are the R symmetry quantum numbers. The 3-point function of these operators (with the normalization of the two point function being set to unity) is given by (also see [48–50]),

$$\langle \mathcal{O}_{\Delta_1,m_1,\bar{m}_1} \mathcal{O}_{\Delta_2,m_2,\bar{m}_2} \mathcal{O}_{\Delta_3,m_3,\bar{m}_3} \rangle = \bar{C}^{(0,0,0)}_{123} \prod_{i<j} \frac{1}{|x_{ij}|^{2\alpha_{ij}}}$$

where, $\alpha_{12} = (\Delta_1 + \Delta_2 - \Delta_3)/2$ etc. and in the large $N$ limit, $\bar{C}^{(0,0,0)}_{123}$ is given by,

$$\bar{C}^{(0,0,0)}_{123} = \frac{(\Delta_1 + \Delta_2 + \Delta_3 + 2)^2}{4\sqrt{N}\ \prod_i(\Delta_i+1)^{1/2}} \frac{\Gamma(\alpha_{13}+1)\,\Gamma(\alpha_{12}+1)}{\Gamma(\Delta_1+1)} \qquad (7.2)$$

Next, let the worldsheet vertex operators "dual" to the boundary CFT chiral primary operators be denoted by $\mathcal{V}_{\Delta,m,\bar{m}}(x,\bar{x};z,\bar{z})$. The vertex operators are related to the boundary CFT operators through (2.1) and, as before, their normalization is given by (E.1). The 3-point function of these vertex operators (in the string theory on $AdS_3 \times S^3 \times T^4$) was computed in [8] and their $\Delta_i$ and $N$ dependence was obtained to be,

$$\bar{F}^{(0,0,0)}_{123} \sim \frac{(\Delta_1 + \Delta_2 + \Delta_3 + 2)^2 \Gamma(\alpha_{13}+1)\,\Gamma(\alpha_{12}+1)}{\sqrt{N\prod_i(\Delta_i+1)}\Gamma(\Delta_1+1)} \prod_i \sqrt{f(\Delta_i)D_0(\Delta_i)} \qquad (7.3)$$

We see that the 3-point function coefficients $\bar{C}^{(0,0,0)}_{123}$ and $\bar{F}^{(0,0,0)}_{123}$ in (7.2) and (7.3) respectively are consistent with the relation (7.1).

Next, we turn our attention to verifying the relation between cubic couplings in bulk supergravity and the OPE coefficients in the boundary theory for the chiral primary operators of $\text{AdS}_3/\text{CFT}_2$ example we are considering. Please note that the straightforward use of Witten diagrams in our approach can only give results that are valid for non-extremal correlators (the bulk cubic couplings of "extremal operators" vanish). In the supergravity approximation, the correlators of CFT chiral primaries and the bulk cubic couplings have been calculated in [51–53]. In this case, there is, however an additional subtlety in identifying the supergravity fields dual to the operators of the boundary CFT. It turns out that none of the chiral primaries mentioned earlier in this section can be identified with

the supergravity scalar fields. A naive matching of operators is inconsistent with the non-renormalization of the correlators. One has to consider a linear combination of these (and other) chiral primaries in the CFT theory if one has to match the non-extremal three point functions computed in the orbifold theory with those computed in the supergravity theory. For extremal correlators, even this is not enough and non-linear operator mixings have to be considered. However we shall not delve deeper into these issues here and we refer the reader to [10, 54] instead.

Here we shall assume that an appropriate linear combination of the chiral primaries in the CFT theory has been taken such that the non-renormalization of the non-extremal three point functions is manifest. We can then compare the relation between the corresponding boundary OPE coefficient and the cubic coupling in supergravity. Following the discussion in [10, 54], we consider a class of scalar supergravity fields $\Sigma_{k,I}$ ($k, I$ being the R symmetry labels). We are going to consider the three point function of this operator. The relevant part of the action is,

$$S = -\frac{N}{4\pi} \int d^3x \sqrt{-G} \left[ \frac{1}{2} \left( \nabla \Sigma_{k,I} \right)^2 + k(k-2)\Sigma_{k,I}^2 + \frac{1}{3!}g_{123}\Sigma_{k_1 I_1}\Sigma_{k_2 I_2}\Sigma_{k_3 I_3} \right] \quad (7.4)$$

It should be noted that $\frac{1}{3!}g_{123}$ is the corresponding coupling constant $U_{123}$ in [10].

Let $\mathcal{O}^{\Sigma}_{\Delta,I}$ be the boundary CFT operator (with dimension $\Delta$) dual to $\Sigma_{k,I}$. We fix the normalization of the 2-point function of these boundary scalar operators to unity. The 3-point function computed using the dual gravity theory is then given by [10, 54],

$$\begin{aligned}
&\left\langle \mathcal{O}^{\Sigma}_{\Delta_1,I_1} \mathcal{O}^{\Sigma}_{\Delta_2,I_2} \mathcal{O}^{\Sigma}_{\Delta_3,I_3} \right\rangle \\
&= \frac{3}{\sqrt{2N}} \frac{g_{123}}{3!} \frac{\Gamma(\alpha_{12})\Gamma(\alpha_{13})\Gamma(\alpha_{23})\Gamma\left(\frac{\Delta_1+\Delta_2+\Delta_3-2}{2}\right)}{\Gamma(\Delta_1)\Gamma(\Delta_2)\Gamma(\Delta_3)} \frac{1}{|x_{12}|^{2\alpha_{12}}|x_{13}|^{2\alpha_{13}}|x_{23}|^{2\alpha_{23}}}
\end{aligned}$$

In section 6, the interaction term comes with a factor $\frac{1}{S_{123}}g_{123}$, and the number of possible Wick contractions cancels with the symmetry factor. In the action (7.4), the interaction terms are defined uniformly (irrespective of the operators in question) and consequently the convention chosen is to multiply by 3! (for every diagram). Furthermore, in order to compare with our calculation, we absorb the overall factor in the action (7.4) into the kinetic term. This means that the cubic coupling that we really need to consider according to our conventions is $\hat{g}_{123} = \frac{2\sqrt{\pi}}{\sqrt{N}}g_{123}$. Doing this, we find the relation between the OPE coefficient $C^{(0,0,0)}_{123}$ and the cubic coupling to be,

$$C^{(0,0,0)}_{123} = \frac{1}{4\sqrt{2}\sqrt{\pi}} \frac{\Gamma(\alpha_{12})\Gamma(\alpha_{13})\Gamma(\alpha_{23})\Gamma\left(\frac{\Delta_1+\Delta_2+\Delta_3-2}{2}\right)}{\Gamma(\Delta_1)\Gamma(\Delta_2)\Gamma(\Delta_3)} \hat{g}_{123}$$

This matches exactly with the relation 6.1 (for $J = 0$ and $d = 2$). The corresponding relation for the three point function of two scalars with the stress tensor (in general dimensions) was verified in [31] itself. It is also straightforward to verify these relations in the $AdS_5/CFT_4$ case using the results presented in [40].

## 8 Discussion

In this paper, we have given an explicit relation between the OPE coefficients in the worldsheet and the boundary theories for the contribution of arbitrary symmetric traceless spin $\ell$ single trace operators to the OPE of single trace scalar operators. We have also considered the case of external conserved vector operators and derived the corresponding relation for the same. Our analysis was done in a model (and dimension) independent manner. The analysis gives a 3-way relation between cubic couplings in the bulk supergravity, boundary OPE coefficients and the worldsheet OPE coefficients for the operators whose 2 and 3-point functions enjoy a non-renormalization theorem. We have also shown that our results are consistent with corresponding known results in the case of $\text{AdS}_3/\text{CFT}_2$ correspondence.

Throughout this paper we have only considered the contribution of single trace operators in the OPE. However in a large $N$ CFT, we also expect to encounter contributions to the OPE from multi trace operators. It is well known that multi trace operators can give rise to logarithmic terms in the OPE due to the anomalous dimensions of such operators. For holographic CFTs, these logarithmic terms can also be seen by analysing the Witten diagrams for corresponding CFT correlation functions in the boundary OPE limit. In the context of $\text{AdS}_3/\text{CFT}_2$ in [7], it was shown that the boundary spacetime interpretation of some of the additional discrete contributions generated during the process of analytic continuation of the worldsheet OPE of normalizable vertex operators could be attributed to multi trace operators in the boundary CFT. However, it was pointed out in [1] that generically not all such additional contributions can be thought as arising due to the exchange of multi trace operators in the spacetime OPE. It is expected that multi trace contributions originate, generically, in a non-local fashion on the worldsheet and a local analysis used in this paper may be insufficient to capture it. It would be very interesting to gain a better understanding of these issues from the perspective of the worldsheet string theory.

We should also mention that our relation between the bulk cubic couplings and the boundary OPE coefficients are not applicable for extremal correlators since the bulk extremal cubic couplings vanish. One needs to use appropriate boundary terms to obtain the corresponding boundary OPE coefficients (see, e.g., [10, 55]).

Apart from understanding the OPE in the boundary CFT from the worldsheet point of view in more detail, it would also be interesting to relate the bootstrap conditions on the boundary and worldsheet theories. This may help in classifying the landscape of large $N$ CFTs which can admit weakly coupled string duals. The explicit relation between the OPE coefficients in the boundary and worldsheet theory will probably be relevant in this context.

### Acknowledgments

We are deeply grateful to Rajesh Gopakumar for suggesting this problem to us, for his help throughout the work and also for his comments on the draft. We would also like to thank Dileep Jatkar and Ashoke Sen for raising various issues which helped in getting

a clearer picture. SS is thankful to Josua Faller for helpful discussions and to Matthias Staudacher for his comments on the draft. SS acknowledges the hospitality and support of the International Center for Theoretical Sciences, Bengaluru. The work of MV was also supported by the SPM research grant of the Council for Scientific and Industrial Research (CSIR), India.

## A   Notations and Conventions

**Notations for Most Used Variables**

1. World-Sheet coordinates $\equiv z, \bar{z}$

2. AdS boundary coordinates $\equiv x, y, \cdots$

3. Boundary space-time operators $\equiv \mathcal{O}_i(x)$

4. World-Sheet Vertex operators $\equiv \mathcal{V}_i(x; z, \bar{z})$

5. Scaling dimension of boundary CFT operators $\equiv \Delta_i$

6. Scaling dimension of world-sheet vertex operators $\equiv h$

7. Quantum number for internal global symmetries $\equiv q$

**Convention for Frequently occuring quantiites**

In the cases where OPE and 3-point functions are charecterized by only one undetermined coefficient (which is always the case in this paper), we shall use the following convetions:

1. The 3-point function coefficient of operators with dimensions $\Delta_1, \Delta_2, \Delta_3$ and spins $\ell_1, \ell_2, \ell_3$ in the boundary CFT is denoted by $\bar{C}_{\Delta_1 \Delta_2 \Delta_3}^{(\ell_1, \ell_2, \ell_3)}$.

2. The OPE coefficient of OPE between two operators of dimensions $\Delta_1, \Delta_2$ and spins $\ell_1, \ell_2$ in the boundary CFT will be denoted by $C_{\Delta_1 \Delta_2 \Delta_3}^{(\ell_1, \ell_2, \ell_3)}$ where $\Delta_3$ and $\ell_3$ denote the dimension and spin of the operator which contributes to the OPE.

3. The 3-point function coefficient of world-sheet vertex operators which are charactrized by boundary space-time scaling dimensions $\Delta_1, \Delta_2, \Delta_3$ and spins $\ell_1, \ell_2, \ell_3$ (under the global symmetry due to boundary CFT) will be denoted by $\bar{F}_{\Delta_1 \Delta_2 \Delta_3}^{(\ell_1, \ell_2, \ell_3)}$.

4. The OPE coefficient of OPE between two world-sheet vertex operators charactrized by boundary scaling dimensions $\Delta_1, \Delta_2$ and spins $\ell_1, \ell_2$ (under global symmetry) will be denoted by $F_{\Delta_1 \Delta_2 \Delta_3}^{(\ell_1, \ell_2, \ell_3)}$ where $\Delta_3$ and $\ell_3$ denote the dimension and spin of the operator which contributes to the OPE.

In all the above, we shall often use just the numbers instead of $\Delta_i$ in the subscript. For example, the 3-point function of two scalars with dimensions $\Delta_1, \Delta_2$ and a spin $\ell$ operator of dimension $\Delta$ will be denoted as $\bar{C}_{12\Delta}^{(0,0,\ell)}$.

**Some Frequently Occuring Functions**

1. The following combination of the coordinates $x_1, x_2$ and $x$ appears frequently in our analysis

$$Z^\mu(x_1, x_2, x) \equiv \frac{(x_1 - x)^\mu}{(x_1 - x)^2} - \frac{(x_2 - x)^\mu}{(x_2 - x)^2} \tag{A.1}$$

We shall always identify it by its arguments whenever this function appears. Note that the position of the co-ordinates in the argument is important for this function.

2. Following product also appears frequently in our expressions

$$\lambda_\ell(a, b) \equiv \prod_{r=0}^{\ell-1} \left( \frac{a+b}{2} + \ell - 1 + r \right) \tag{A.2}$$

It is symmetric in the arguments $a$ and $b$.

3. Finally, we define

$$J^{\mu\nu}(x) \equiv \delta^{\mu\nu} - 2\frac{x^\mu x^\nu}{x^2} \tag{A.3}$$

# B  Some Useful CFT Correlators

In this appendix, we note down some CFT correlators which we shall need in this paper.

1. The two point function of spin-$\ell$ symmetric traceless operators with conformal dimension $\Delta$ is given by

$$\langle \mathcal{O}^{\mu_1 \cdots \mu_\ell}(x) \mathcal{O}_{\nu_1 \cdots \nu_\ell}(0) \rangle = K_\ell(\Delta) \left[ \frac{J^{\mu_1}_{(\nu_1}(x) \cdots J^{\mu_\ell}_{\nu_\ell)}(x) - \text{Traces}}{|x|^{2\Delta}} \right] \tag{B.1}$$

$K_\ell(\Delta)$ is the two point function coefficient for spin $\ell$ operators and $J_{\mu\nu}(x)$ is defined in (A.3). We shall normalise the scalar operators to unity so that $K_0(\Delta) = 1$. However, for $\ell \neq 0$, we shall keep the explicit factors of $K_\ell(\Delta)$.

2. The 3-point function of two scalars and one spin $\ell$ operator is given by

$$\langle \mathcal{O}_1(x_1)\mathcal{O}_2(x_2)\mathcal{O}_3^{\mu_1\cdots\mu_\ell}(x_3) \rangle = \frac{\bar{C}_{123}^{(0,0,\ell)} \left( Z^{\mu_1}(x_1, x_2, x_3) \cdots Z^{\mu_\ell}(x_1, x_2, x_3) - \text{traces} \right)}{|x_{12}|^{\Delta_1+\Delta_2-\Delta_3+\ell}|x_{23}|^{\Delta_2+\Delta_3-\Delta_1-\ell}|x_{13}|^{\Delta_1+\Delta_3-\Delta_2-\ell}} \tag{B.2}$$

$Z^\mu(x_1, x_2, x_3)$ is defined in equation (A.1).

3. An operator $\mathcal{O}_\Delta^{\mu_1\cdots\mu_\ell}$ represents a conserved current if $\Delta = \ell + d - 2$. In particular, for global symmetry currents ($\ell = 1, \Delta_1 = \Delta_2 = d-1$), we have the following result [56]

$$\langle \mathcal{O}^\mu(x_1)\mathcal{O}^\nu(x_2)\mathcal{O}_{\Delta_3}(x_3) \rangle = \bar{C}_{123}^{(1,1,0)} \frac{J^\mu_{\ \sigma}(x_1 - x_3)J^{\sigma\nu}(x_3 - x_2) + c\ J^{\mu\nu}(x_1 - x_2)}{|x_{12}|^{2(d-1)-\Delta_3}|x_{23}|^{\Delta_3}|x_{13}|^{\Delta_3}} \tag{B.3}$$

where, the constant $c$ is given by

$$c = \frac{\Delta_3 - 2(d-1)}{\Delta_3} \tag{B.4}$$

## C    Some Useful OPEs

In this appendix, we note down some of the boundary and world-sheet OPEs which we shall need in our calculations.

### C.1    Scalar-Scalar Boundary CFT OPE

The OPE of two scalar operators in a boundary CFT has the following general structure

$$\mathcal{O}_1(x_1)\mathcal{O}_2(x_2) = \sum_{\Delta,q} C_{12\Delta}^{(0,0,\ell)}(\Delta_i; q_i, q) \frac{(x_{12})_{\mu_1}\cdots(x_{12})_{\mu_\ell}}{|x_1 - x_2|^{\Delta_1+\Delta_2-\Delta+\ell}} \, \mathcal{O}_{\Delta,q}^{\mu_1\cdots\mu_\ell}(x_2) + \cdots \quad \text{(C.1)}$$

$\mathcal{O}_{\Delta,q}^{\mu_1\cdots\mu_\ell}$ is a primary operator of dimension $\Delta$. $q$ denotes the internal quantum numbers coming from some possible global symmetries of the boundary theory. The $\cdots$ terms denote the contributions due to the descendants and the multi trace operators.

### C.2    Vector-Vector Boundary CFT OPE

We shall be interested in the contribution of scalars to the OPE of two vector operators. By rotation invariance, this contribution can have the following general structure (ignoring the descendant and multi trace contributions)

$$\mathcal{O}_1^\mu(x_1)\mathcal{O}_2^\nu(0) = \sum_\Delta C_{12\Delta}^{(1,1,0)} \frac{\hat{x}_1^\mu\hat{x}_1^\nu + b\,\delta^{\mu\nu}}{|x_1|^{\Delta_1+\Delta_2-\Delta}}\mathcal{O}_\Delta(0) + \cdots \quad \text{(C.2)}$$

Since in this paper, we are only interested in the conserved currents, the constant $b$ can be fixed by the conformal symmetry. To do this, we shall evaluate the contribution to three point correlator of two spin 1 and one scalar operator in two different ways in the OPE limit. Using the OPE (C.2) for identical spin 1 conserved currents, we obtain

$$\langle\mathcal{O}^\mu(x_1)\mathcal{O}^\nu(0)\mathcal{O}(x_3)\rangle = \sum_\Delta C_{12\Delta}^{(1,1,0)} \frac{\hat{x}_1^\mu\hat{x}_1^\nu + b\,\delta^{\mu\nu}}{|x_1|^{2(d-1)-\Delta}}\langle\mathcal{O}_\Delta(0)\mathcal{O}(x_3)\rangle$$

$$= C_{123}^{(1,1,0)} \frac{\hat{x}_1^\mu\hat{x}_1^\nu + b\,\delta^{\mu\nu}}{|x_1|^{2(d-1)-\Delta_3}|x_3|^{2\Delta_3}} \quad \text{(C.3)}$$

The exact expression of the same correlator as given in (B.3) reduces in the OPE limit $x_1 \to x_2 = 0$ to

$$\langle\mathcal{O}^\mu(x_1)\mathcal{O}^\nu(0)\mathcal{O}(_3)\rangle = \bar{C}_{123}^{(1,1,0)} \frac{\hat{x}_1^\mu\hat{x}_1^\nu - \frac{(1+c)}{2c}\delta^{\mu\nu}}{|x_1|^{2(d-1)-\Delta_3}|x_3|^{2\Delta_3}} \quad \text{(C.4)}$$

comparing (C.3), (C.4) and using (B.4), we obtain

$$b = -\frac{(\Delta + 1 - d)}{\Delta + 2 - 2d} \quad \text{and} \quad \bar{C}_{123}^{(1,1,0)} = C_{123}^{(1,1,0)} \quad \text{(C.5)}$$

## C.3 "Vector-Vector" Worldsheet OPE

We can have operators on the worldsheet which can carry boundary co-ordinate indices. The integral over the worldsheet co-ordinate of these operators can be interpreted as boundary tensor operators. We shall consider two such "vector" operators and focus on the contribution of the scalar operators in their OPE. The most general form of this contribution can be written as

$$\mathcal{V}_1^\mu(x_1, z_1)\mathcal{V}_2^\nu(x_2, 0) = \sum_q \int_C d\Delta \int d^d x \; F^{\mu\nu}(z_1; x_i, x; \Delta_i, \Delta; q_i, q)\mathcal{V}_{\Delta,q}(x, 0)$$

where, as usual, we have ignored the contribution of descendants.

The $x$ and $z$ dependence of $F^{\mu\nu}$ is fixed by the conformal invariance. Using the tensor structure from the 3-point function of two vectors and one scalar as given in equation (B.3), we can write

$$\mathcal{V}_1^\mu(x_1, z_1)\mathcal{V}_2^\nu(x_2, 0)$$
$$= \sum_q \int_C d\Delta \int d^d x \; \frac{F_{12\Delta}^{(1,1,0)}}{|z_1|^{h_1+h_2-h_{\Delta,q}}} \frac{\left(\bar{c}J^{\mu\nu}(x_1-x_2) + J^\mu{}_\sigma(x_1-x)J^{\sigma\nu}(x_2-x)\right)}{|x_1-x_2|^\alpha|x_2-x|^\beta|x_1-x|^\gamma}\mathcal{V}_{\Delta,q}(x, 0)$$

By demanding the OPE relation to be invariant under conformal transformation or using the shadow operator trick described in section (3.1), we obtain the functional forms of $\alpha, \beta$ and $\gamma$ in this case to be

$$\alpha = 2(d-1) + \Delta - d \quad , \quad \beta = d - \Delta = \gamma \tag{C.6}$$

Now, setting $x_2 = 0$ and making the coordinate transformation $x = y|x_1|$, we obtain

$$\mathcal{V}_1^\mu(x_1, z_1)\mathcal{V}_2^\nu(0, 0)$$
$$= \sum_q \int_C d\Delta \; F_{12\Delta}^{(1,1,0)} \frac{|z_1|^{-(h_1+h_2-h_{\Delta,q})}}{|x_1|^{\alpha+\beta+\gamma-d}} \int d^d y \; \frac{1}{y^\beta \, |y-\hat{x}_1|^\gamma}$$
$$\left[\bar{c}\left(\delta^{\mu\nu} - 2\hat{x}_1^\mu\hat{x}_1^\nu\right) + \left(\delta^\mu{}_\sigma - 2\frac{(\hat{x}_1-y)^\mu(\hat{x}_1-y)_\sigma}{(\hat{x}_1-y)^2}\right)\left(\delta^{\sigma\nu} - 2\frac{y^\sigma y^\nu}{y^2}\right)\right]\mathcal{V}_{\Delta,q}(y|x_1|, 0)$$

We are interested in the small $|x_1|$ limit and we are also ignoring the contributions of the descendants. So, as before, we expand the $\mathcal{V}_{\Delta,q}^\mu$ for small $x_1$ and keep only the first term in its Taylor expansion to get

$$\mathcal{V}_1^\mu(x_1, z_1)\mathcal{V}_2^\nu(0, 0) = \sum_q \int_C d\Delta \frac{|z_1|^{-(h(1)+h(2)-h(\Delta,q))}}{|x_1|^{\alpha+\beta+\gamma-d}} F_{12\Delta}^{(1,1,0)} \; \mathcal{V}_{\Delta,q}(0, 0) \; H^{\mu\nu}(x_1) \text{ (C.7)}$$

where, using the integrals given in the appendix (G), we have

$$
\begin{aligned}
&H^{\mu\nu} \\
&\equiv \int \frac{d^d y}{|y|^\beta \, |y - \hat{x}_1|^\gamma} \left[ \bar{c} \left( \delta^{\mu\nu} - 2\hat{x}_1^\mu \hat{x}_1^\nu \right) + \left( \delta^\mu_{\ \sigma} - 2\frac{(\hat{x}_1 - y)^\mu (\hat{x}_1 - y)^\sigma}{(\hat{x}_1 - y)^2} \right) \left( \delta^{\sigma\nu} - 2\frac{y^\sigma y^\nu}{y^2} \right) \right] \\
&= \left[ \left\{ \bar{c} + \left( \frac{\beta(\gamma - 2) - 2\gamma + 2d}{\beta\gamma} \right) \right\} \delta^{\mu\nu} - \left\{ 2\bar{c} + \frac{2(d-2)(\beta + \gamma - d)}{\beta\gamma} \right\} \hat{x}_1^\mu \hat{x}_1^\nu \right] \\
&\quad \frac{\pi^{d/2} \Gamma\left( \frac{d-\beta}{2} \right) \Gamma\left( \frac{d-\gamma}{2} \right) \Gamma\left( \frac{\beta+\gamma-d}{2} \right)}{\Gamma\left( \frac{\beta}{2} \right) \Gamma\left( \frac{\gamma}{2} \right) \Gamma\left( \frac{2d-\beta-\gamma}{2} \right)}
\end{aligned}
\tag{C.8}
$$

Since we are considering the conserved currents, the constant $\bar{c}$ is fixed and is given by replacing $\Delta$ with $d - \Delta$ in equation (B.4) to be

$$
\bar{c} = \frac{2 - d - \Delta}{d - \Delta}
$$

Using (C.6), we thus obtain

$$
H^{\mu\nu} = \frac{\pi^{d/2} \Gamma\left( \frac{\Delta}{2} \right) \Gamma\left( \frac{\Delta}{2} \right) \Gamma\left( \frac{d-2\Delta}{2} \right)}{\Gamma\left( \frac{d-\Delta}{2} \right) \Gamma\left( \frac{d-\Delta}{2} \right) \Gamma\left( \Delta \right)} \left[ \frac{-2\Delta(\Delta + 2 - 2d)}{(d-\Delta)^2 (\Delta + 1 - d)} \right] \left[ \hat{x}^\mu \hat{x}^\nu - \frac{(\Delta + 1 - d)}{(\Delta + 2 - 2d)} \delta^{\mu\nu} \right]
$$

Note that the ratio of the coefficients of $\delta^{\mu\nu}$ and $\hat{x}_1^\mu \hat{x}_1^\nu$ is precisely the corresponding ratio (C.5) of the boundary CFT calculation.

## D  Relation Between OPE Coefficients and 3-point Function Coefficients

In our calculations, we need the relation between the 3-point function coefficients and the OPE coefficients for the boundary CFT theory and the world-sheet theory. We shall consider each case below. We shall be interested in obtaining the relationship for three point function of two scalars and one spin $\ell$ operator and the OPE coefficient for spin $\ell$ exchange between the OPE of two scalar operators.

### D.1  Boundary CFT Theory

To obtain the relation between the 3-point function coefficient of two scalar and one spin $\ell$ operator, namely $\bar{C}_{123}^{(0,0,\ell)}$ and the OPE coefficient $C_{123}^{(0,0,\ell)}$ (i.e. spin $\ell$ contribution to the OPE of two scalars), we shall evaluate the 3-point function of two scalars and one spin $\ell$ operator in the OPE limit using two different ways. Equating the two expressions will give us the desired result.

We first evaluate the 3-point function by using the OPE method. Considering spin $\ell$ exchange in the OPE and using (B.1) gives,

$$
\begin{aligned}
&\left\langle O_1(x_1) O_2(0) O_3^{\mu_1 \cdots \mu_\ell}(x_3) \right\rangle \\
&= \sum_{\Delta, q} C_{12\Delta}^{(0,0,\ell)}(q_1, q_2; q) \frac{(x_1)^{\nu_1} \cdots (x_1)^{\nu_\ell}}{|x_1|^{\Delta_1 + \Delta_2 - \Delta + \ell}} \left\langle O_{\nu_1 \cdots \nu_\ell \Delta, q}(0) \mathcal{O}_3^{\mu_1 \cdots \mu_\ell}(x_3) \right\rangle \\
&= K_\ell(\Delta_3) C_{123}^{(0,0,\ell)}(q_1, q_2; q) \frac{(x_1)^{\nu_1} \cdots (x_1)^{\nu_\ell}}{|x_1|^{\Delta_1 + \Delta_2 - \Delta_3 + \ell}} \left[ \frac{J^{\mu_1}_{(\nu_1}(\hat{x}_3) \cdots J^{\mu_\ell}_{\nu_\ell)}(\hat{x}_3) - \text{Traces}}{|x_3|^{2\Delta_3}} \right]
\end{aligned}
$$

Alternatively, the exact expression of the 3-point function of two scalars and one spin $\ell$ operator fixed by the conformal invariance is given in equation (B.2). Setting $x_2 = 0$ in (B.2) and evaluating it in the small $x_1$ limit gives an expression whose $x$ dependence matches precisely with the corresponding $x$ dependence of the result obtained using the OPE method above. This gives,

$$\bar{C}_{123}^{(0,0,\ell)} = K_\ell(\Delta_3)C_{123}^{(0,0,\ell)} \tag{D.1}$$

## D.2 World-sheet Theory

For the world-sheet theory, we want to obtain the relationship between the 3-point function coefficient $\bar{F}_{123}^{(0,0,\ell)}$ and the OPE coefficient $F_{123}^{(0,0,\ell)}$. We again consider the three point function of one spin $\ell$ and two spin zero vertex operators. Using the world sheet OPE (3.9) between the vertex operators $\mathcal{V}_1(x_1)$ and $\mathcal{V}_2(x_2)$, this 3-point correlator can be evaluated as

$$\langle \mathcal{V}_1(x_1, z_1)\mathcal{V}_2(0,0)\mathcal{V}_{\Delta_3,q_3}^{\mu_1\cdots\mu_\ell}(x_3, z_3)\rangle$$
$$= \sum_q \int_C d\Delta \frac{|z_1|^{-(h(1)+h(2)-h(\Delta,q))}}{|x_1|^{\alpha+\beta+\gamma-D+\ell}} F_{12\Delta}^{(0,0,\ell)}\langle \mathcal{V}_{\Delta,q}^{\nu_1\cdots\nu_\ell}(0,0)\mathcal{V}_{\Delta_3,q_3}^{\mu_1\cdots\mu_\ell}(x_3, z_3)\rangle G_{\nu_1\cdots\nu_\ell}(x_1)$$

By conformal invariance, the 2-point function of two spin $\ell$ vertex operators on the world-sheet can be written as

$$\langle \mathcal{V}_1^{\mu_1\cdots\mu_\ell}(x_1, z_1)\mathcal{V}_{2\,\nu_1\cdots\nu_\ell}(x_2, z_2)\rangle = \frac{\delta_{q_1,q_2}}{|z_{12}|^{2h_1}}\left[D_\ell(\Delta_1)\delta(\Delta_1 - \Delta_2)\frac{J_{\nu_1}^{(\mu_1}\cdots J_{\nu_\ell}^{\mu_\ell)} - \mathrm{Traces}}{|x_{12}|^{2\Delta_1}}\right] \tag{D.2}$$

$D_\ell(\Delta)$ is a constant depending upon the conformal dimension. Using the above expression, we obtain

$$\langle \mathcal{V}_1(x_1, z_1)\mathcal{V}_2(0,0)\mathcal{V}_{\Delta_3,q_3}^{\mu_1\cdots\mu_\ell}(x_3; z_3)\rangle$$
$$= F_{123}^{(0,0,\ell)}D_\ell(\Delta_3)\frac{|z_1|^{-(h(1)+h(2)-h(\Delta_3,q_3))}}{|x_1|^{\alpha+\beta+\gamma-D+\ell}}\frac{\left(J_{(\nu_1}^{\mu_1}(x_3)\cdots J_{\nu_\ell)}^{\mu_\ell}(x_3) - \mathrm{Traces}\right)}{|z_3|^{2h_3}\ |x_3|^{2\Delta_3}}G^{\nu_1\cdots\nu_\ell}(x_1)$$

Alternatively, the universal expression for the same correlator is fixed by the conformal invariance to be

$$\langle \mathcal{V}_1(x_1; z_1)\mathcal{V}_2(x_2; z_2)\mathcal{V}_3^{\mu_1\cdots\mu_\ell}(x_3; z_3)\rangle = \frac{\bar{F}_{123}^{(0,0,\ell)}}{|z_{12}|^{h_1+h_2-h_3}|z_{23}|^{h_2+h_3-h_1}|z_{31}|^{h_3+h_1-h_2}}$$
$$\times \frac{\left(Z^{\mu_1}(x_1, x_2, x_3)\cdots Z^{\mu_\ell}(x_1, x_2, x_3) - \mathrm{Traces}\right)}{|x_{12}|^{\Delta_1+\Delta_2-\Delta_3+\ell}|x_{23}|^{\Delta_2+\Delta_3-\Delta_1-\ell}|x_{31}|^{\Delta_3+\Delta_1-\Delta_2-\ell}} \tag{D.3}$$

For our case, this reduces to

$$\langle \mathcal{V}_1(x_1; z_1)\mathcal{V}_4(0;0)\mathcal{V}_{\Delta_3,q_3}^{\mu_1\cdots\mu_\ell}(x_3; z_3)\rangle$$
$$= \frac{\bar{F}_{123}^{(0,0,\ell)}}{|z_1|^{h_1+h_1-h_3}|z_3|^{2h_3}}\frac{\left(Z^{\mu_1}(x_1, 0, x_3)\cdots Z^{\mu_\ell}(x_1, 0, x_3) - \mathrm{Traces}\right)}{|x_1|^{\Delta_1+\Delta_2-\Delta_3+\ell}|x_3|^{2(\Delta_3-\ell)}}$$

Simplifying the tensor structure in numerator of the above expression, comparing with the result obtained using the OPE method and using (3.10) , we obtain the desired result

$$\bar{F}_{123}^{(0,0,\ell)} = \left[ \pi^{d/2} \lambda_\ell(\beta,\gamma) D_\ell(\Delta_3) \frac{\Gamma\left(\frac{d-\beta}{2}\right) \Gamma\left(\frac{d-\gamma}{2}\right) \Gamma\left(\frac{\beta+\gamma-d}{2} + \ell\right)}{\Gamma\left(\frac{\beta}{2} + \ell\right) \Gamma\left(\frac{\gamma}{2} + \ell\right) \Gamma\left(\frac{2d-\beta-\gamma}{2}\right)} \right] F_{123}^{(0,0,\ell)} \tag{D.4}$$

where $\lambda_\ell(\beta,\gamma)$ is defined in equation (A.2).

## E   Normalization of World-Sheet Vertex Operators

In order to match correlators computed using the worldsheet theory with the correlators computed in the boundary CFT theory, we need to fix the relative normalization between the world-sheet vertex operators and the boundary CFT operators. A convenient way to do this is to fix the normalization of the two point functions. We have chosen the normalization of the boundary CFT two point functions of spin $\ell$ operators to be $K_\ell(\Delta)$ with $K_0(\Delta) = 1$. By computing the boundary CFT two point functions using the world-sheet theory and demanding its normalization to be $K_\ell(\Delta)$ fixes the relative normalization of the vertex operators.

The two point function of two vertex operators (not normalized yet) is given by (D.2). Fixing the world-sheet coordinates at $z = 0$ and $z = 1$ in this, the boundary CFT two point function can be computed as follows,

$$\left\langle \mathcal{O}_1^{\mu_1\cdots\mu_\ell}(x_1)\mathcal{O}_{2\nu_1\cdots\nu_\ell}(x_2) \right\rangle = \frac{1}{V_{\text{conf}}} \left\langle \mathcal{V}_1^{\mu_1\cdots\mu_\ell}(x_1, z_1 = 1)\mathcal{V}_{2\nu_1\cdots\nu_\ell}(x_2, z_2 = 0) \right\rangle$$
$$= f(\Delta) \left[ \frac{D_\ell(\Delta)\left(J_{\nu_1}^{(\mu_1} \cdots J_{\nu_\ell}^{\mu_\ell)} - \text{Traces}\right)}{|x_{12}|^{2\Delta}} \right]$$

We have divided the right hand side by the volume of the conformal group $V_{\text{conf}}$ on sphere since we have fixed the world-sheet coordinates. This factor cancels the divergence arising due to the delta function in equation (D.2) upto a factor $f(\Delta)$. In the situations where the scaling dimensions and the spectrum of the theory is explicitly known such as in the case of $AdS_3/CFT_2$ correspondence, this factor can be determined explicitly (see, e.g., [8]).

To match the above result with (B.1), the relative normalization between the boundary and the world-sheet vertex operators should be fixed as

$$\mathcal{O}_\Delta(x) = \frac{1}{a_\Delta^{(\ell)}} \int d^2z \; \mathcal{V}_\Delta(x, z) \tag{E.1}$$

where,

$$a_\Delta^{(\ell)} \equiv \sqrt{\frac{f(\Delta)D_\ell(\Delta)}{K_\ell(\Delta)}} \tag{E.2}$$

## F  Alternative Way of Fixing the World-Sheet OPE Structure

As mentioned in section 3.2, the tensor structure of the OPE (3.5) can be fixed by demanding the conformal invariance and noting the 3-point correlator of two scalars and one spin $\ell$ object as given in equation (B.2). This would give the same result as (3.7) upto the powers $\alpha, \beta$ and $\gamma$ in the denominator. These powers can be determined by demanding the OPE relation (3.7) to be invariant under the conformal transformation. We now show that the action of dilatation and inversion on the OPE relation (3.7) gives the same result as given in (3.8).

If the vertex operator $\mathcal{V}$ has scaling dimension $\Delta$ from the boundary CFT point of view, then under dilatation $x \to x' = \lambda x$, it transforms as

$$\mathcal{V}(x, z) \to \lambda^{-\Delta} \, \mathcal{V}(x, z)$$

and hence, the OPE relation (3.7) becomes

$$
\lambda^{-\Delta_1 - \Delta_2} \mathcal{V}_1(x_1, z_1) \mathcal{V}_2(x_2, 0)
$$
$$
= \sum_q \int_C d\Delta \, (\lambda)^{-\alpha - \beta - \gamma + d - \Delta - \ell} \int d^d x \, \frac{|z_1|^{-(h_1 + h_2 - h_{\Delta,q})} Z^{\mu_1} \cdots Z^{\mu_\ell}}{|x_1 - x_2|^\alpha |x_2 - x|^\beta |x_1 - x|^\gamma} F^{(0,0,\ell)}_{12\Delta} \mathcal{V}^{\mu_1 \cdots \mu_\ell}_{\Delta,q}(x, 0)
$$

where, $Z^\mu = Z^\mu(x_1, x_2, x)$ and the factor of $\lambda^{-\ell}$ in the right hand side comes from the terms involving $Z^\mu$ in the numerator.

Comparing the two sides of the above equation, we get,

$$\alpha + \beta + \gamma = \Delta_1 + \Delta_2 - \Delta + d - \ell \qquad (\text{F.1})$$

Next, we consider the effect of inversion on the boundary coordinates $x_i$ in (3.7) in which we have

$$
x^\mu \to \frac{x^\mu}{x^2} \quad , \quad x^2_{ij} \to \frac{x^2_{ij}}{x^2_i x^2_j} \quad , \quad d^d x \to \frac{d^d x}{|x|^{2d}} \quad , \quad \mathcal{V}(x, z) \to |x|^{2\Delta} \mathcal{V}(x, z)
$$

$$
\mathcal{V}^{\mu_1 \cdots \mu_\ell}(x, z) \to |x|^{2\Delta} J^{\mu_1}_{\nu_1}(x) \cdots J^{\mu_\ell}_{\nu_\ell}(x) \mathcal{V}^{\nu_1 \cdots \nu_\ell}(x, z)
$$

We also have [6]

$$
(Z^{\mu_1} \cdots Z^{\mu_\ell}) \mathcal{V}_{\mu_1 \cdots \mu_\ell}(x, z) \to |x|^{2\Delta + 2\ell} (Z^{\mu_1} \cdots Z^{\mu_\ell}) \mathcal{V}_{\mu_1 \cdots \mu_\ell}(x, z)
$$

Using these, the OPE (3.7) gives

$$
x_1^{2\Delta_1} x_2^{2\Delta_2} \mathcal{V}_1(x_1, z_1) \mathcal{V}_2(x_2, 0)
$$
$$
= \sum_q \int_C d\Delta \int d^d x \, \frac{|z_1|^{-(h(1) + h(2) - h(\Delta, q))}}{|x_{12}|^\alpha |x_2 - x|^\beta |x - x_1|^\gamma} |x_1|^{\alpha + \gamma} |x_2|^{\alpha + \beta} |x|^{\beta + \gamma - 2d + 2\Delta + 2\ell}
$$
$$
Z_{\mu_1}(x_1, x_2, x) \cdots Z_{\mu_\ell}(x_1, x_2, x) F^{(0,0,\ell)}_{12\Delta} \mathcal{V}^{\mu_1 \cdots \mu_\ell}_{\Delta,q}(x, 0)
$$

---

[6]To see this for spin 1, we note that

$$
\left[ \frac{x_1^\mu x_3^2 - x_3^\mu x_1^2}{x_{13}^2} - \frac{x_2^\mu x_3^2 - x_3^\mu x_2^2}{x_{23}^2} \right] \left( \delta_{\mu\nu} - 2 \frac{x_{3\mu} x_{3\nu}}{x_3^2} \right) = \left( \frac{x_{13}^\mu}{x_{13}^2} - \frac{x_{23}^\mu}{x_{23}^2} \right) (x_3)^2
$$

Comparing both sides gives us the desired result

$$\alpha = \Delta_1 + \Delta_2 + \Delta - d + \ell,$$
$$\beta = \Delta_2 - \Delta_1 - \Delta + d - \ell,$$
$$\gamma = \Delta_1 - \Delta_2 - \Delta + d - \ell$$

It should be noted that this solution automatically satisfies the constraint (F.1).

## G  Some Useful Integrals

In this appendix, we shall note down some integrals which occur frequently in our calculations. The references [57, 58] are helpful in obtaining some of these integrals.

1. Following basic integral is the building block of the more complicated integrals which we shall need in this paper

$$I(a,b) \equiv \int \frac{d^d y}{|y|^a \, |y - \hat{x}|^b} = \frac{\pi^{d/2} \Gamma\left(\frac{d-a}{2}\right) \Gamma\left(\frac{d-b}{2}\right) \Gamma\left(\frac{a+b-d}{2}\right)}{\Gamma\left(\frac{a}{2}\right) \Gamma\left(\frac{b}{2}\right) \Gamma\left(\frac{2d-a-b}{2}\right)} \tag{G.1}$$

Note that this is manifestly symmetric in $a$ and $b$.

2. Another useful integral is

$$\int \frac{d^d y}{|y_1 - y|^{a_1} |y_2 - y|^{a_2} |y_3 - y|^{a_3}}$$
$$= \frac{\pi^{d/2}}{|y_{12}|^{d-a_3} |y_{23}|^{d-a_1} |y_{13}|^{d-a_2}} \frac{\Gamma\left(\frac{d-a_1}{2}\right) \Gamma\left(\frac{d-a_2}{2}\right) \Gamma\left(\frac{d-a_3}{2}\right)}{\Gamma\left(\frac{a_1}{2}\right) \Gamma\left(\frac{a_2}{2}\right) \Gamma\left(\frac{a_3}{2}\right)} \tag{G.2}$$

3. Next, we consider

$$\int d^d y \, \frac{y^\mu}{|y|^a \, |y - \hat{x}|^b} = \frac{\pi^{d/2} \Gamma\left(\frac{d-a+2}{2}\right) \Gamma\left(\frac{d-b}{2}\right) \Gamma\left(\frac{a+b-d}{2}\right)}{\Gamma\left(\frac{a}{2}\right) \Gamma\left(\frac{b}{2}\right) \Gamma\left(\frac{2d-a-b+2}{2}\right)} \hat{x}^\mu$$

To evaluate this integral, we note that it can only be proportional to $\hat{x}^\mu$, since there is no other vector in the problem. Hence, we write $I^\mu(a,b) = h(a,b,x^2)\hat{x}^\mu$. Now, contracting both sides with $\hat{x}^\mu$, completing the square and using (G.1) gives the desired result.

4. We shall also need

$$\int d^d y \, \frac{y^\mu y^\nu}{|y|^a \, |y - \hat{x}|^b}$$
$$= \pi^{d/2} \left[ \frac{\Gamma\left(\frac{d-a+4}{2}\right) \Gamma\left(\frac{d-b}{2}\right) \Gamma\left(\frac{a+b-d}{2}\right)}{\Gamma\left(\frac{a}{2}\right) \Gamma\left(\frac{b}{2}\right) \Gamma\left(\frac{2d-a-b+4}{2}\right)} \hat{x}^\mu \hat{x}^\nu + \frac{\Gamma\left(\frac{d-a+2}{2}\right) \Gamma\left(\frac{d-b+2}{2}\right) \Gamma\left(\frac{a+b-d-2}{2}\right)}{2\Gamma\left(\frac{a}{2}\right) \Gamma\left(\frac{b}{2}\right) \Gamma\left(\frac{2d-a-b+4}{2}\right)} \eta^{\mu\nu} \right]$$

To evaluate this integral, we note that the Lorentz invariance allows this integral to be a linear combination of terms proportional to $\hat{x}^\mu \hat{x}^\nu$ and $\eta^{\mu\nu}$. Hence, we can write

$$\int d^d y \, \frac{y^\mu y^\nu}{|y|^a \, |y - \hat{x}|^b} = g_1(a,b)\hat{x}^\mu \hat{x}^\nu + g_2(a,b)\eta^{\mu\nu}$$

Contracting it with $\eta_{\mu\nu}$ and $\hat{x}_\mu\hat{x}_\nu$ and solving the resulting equations, we obtain the desired result.

5. For calculations involving world-sheet vectors, we need

$$I^{\mu\nu\lambda}(a,b) \equiv \int d^d y \, \frac{y^\mu y^\nu y^\lambda}{|y|^a \, |y-\hat{x}|^b}$$

To evaluate this integral, we note that the Lorentz invariance allows it to be a linear combination of terms made up of $\hat{x}^\mu, \hat{x}^\nu, \hat{x}^\lambda$ and $\eta^{\mu\nu}$. Hence, we can write

$$I^{\mu\nu\lambda}(a,b) = f_1(a,b)\hat{x}^\mu\hat{x}^\nu\hat{x}^\lambda + f_2(a,b)\left(\eta^{\mu\nu}\hat{x}^\lambda + \eta^{\mu\lambda}\hat{x}^\nu + \eta^{\lambda\nu}\hat{x}^\mu\right)$$

The last 3 terms have same coefficients since $I^{\mu\nu\lambda}$ is symmetric in all three indices. Again, Contracting it with various combinations of $\hat{x}^\mu, \hat{x}^\nu, \hat{x}^\lambda$ and $\eta^{\mu\nu}$ and solving the resulting equations gives

$$f_1 = \frac{\pi^{d/2}\Gamma\left(\frac{a+b-d}{2}\right)\Gamma\left(\frac{d-a+6}{2}\right)\Gamma\left(\frac{d-b}{2}\right)}{\Gamma\left(\frac{a}{2}\right)\Gamma\left(\frac{b}{2}\right)\Gamma\left(3+d-\frac{a+b}{2}\right)}$$

$$f_2 = \frac{\pi^{d/2}\Gamma\left(\frac{a+b-d-2}{2}\right)\Gamma\left(\frac{d-a+4}{2}\right)\Gamma\left(\frac{d-b+2}{2}\right)}{2\Gamma\left(\frac{a}{2}\right)\Gamma\left(\frac{b}{2}\right)\Gamma\left(3+d-\frac{a+b}{2}\right)}$$

6. For calculations involving spin $\ell$ operators, we shall encounter the following integral (2.19 of [59])

$$\int d^d y \, \frac{J^{\mu_1}_{(\nu_1}(x-y)\cdots J^{\mu_\ell}_{\nu_\ell)}(x-y)Z^{\nu_1}(x_1,x_2,y)\cdots Z^{\nu_\ell}(x_1,x_2,y)}{|x_1-y|^a|x_2-y|^b|x-y|^c}$$
$$= \pi^{d/2}\frac{\lambda_\ell(a,b)\Gamma\left(\frac{d-a}{2}\right)\Gamma\left(\frac{d-b}{2}\right)\Gamma\left(\frac{d-c}{2}\right)}{\Gamma\left(\frac{a}{2}+\ell\right)\Gamma\left(\frac{b}{2}+\ell\right)\Gamma\left(\frac{c}{2}+\ell\right)}\frac{Z^{\mu_1}(x_1,x_2,x)\cdots Z^{\mu_\ell}(x_1,x_2,x)}{|x-x_1|^{a+c-d}|x-x_2|^{b+c-d}|x_1-x_2|^{d-c}}$$
$$\tag{G.3}$$

where, $a+b+c = 2d-2\ell$.

7. Another useful integral which is related to the integral (G.3) is (2.19 of [59])

$$\int d^d y \, \frac{Z_{\mu_1}\cdots Z_{\mu_\ell}}{|x_1-y|^a|x_2-y|^b} = \pi^{d/2}\lambda_\ell\frac{\Gamma\left(\frac{a+b-d}{2}+\ell\right)\Gamma\left(\frac{d-a}{2}\right)\Gamma\left(\frac{d-b}{2}\right)}{\Gamma\left(d-\frac{a+b}{2}\right)\Gamma\left(\frac{a}{2}+\ell\right)\Gamma\left(\frac{b}{2}+\ell\right)}\frac{(x_{12})_{\mu_1}\cdots(x_{12})_{\mu_\ell}}{|x_{12}|^{2\left(\frac{a+b-d}{2}+\ell\right)}}$$
$$\tag{G.4}$$

where, $Z^\mu = Z^\mu(x_1,x_2,y)$ in the integrand.

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
