# Peer review of "Implications of the AdS/CFT Correspondence on Spacetime and Worldsheet OPE Coefficients"

_SciPost Physics Core_

## Round 2 · Referee Report · Anonymous (Referee 1) · 2023-11-25

Report

This paper studied the OPEs of vertex operators in a string worldsheet theory in AdS space. Under the AdS/CFT correspondence, the vertex operators in the worldsheet CFT correspond to single trace operators in the dual boundary CFT. The main results include:

  1. Derivation of relations between the OPE coefficients in the worldsheet CFT and those in the dual boundary CFT, extending the results of reference [1] to spinning operators.

  2. Verification of these results against the worldsheet correlators in the AdS3/CFT2 correspondence in reference [8].

This paper is well-written and presents new results. I recommend it for publication.

---

## Round 2 · Referee Report · Anonymous (Referee 2) · 2023-12-27

Strengths

The authors extend the analysis of the reference $[1]$ of the manuscript for the spinning case. The manuscript, as a result, shares some of the technical subtleties, appearing in the reference $1$. The extension from the scalar case to the spinning case is straightforward as already pointed out in the reference $[1]$, in the beginning of the section $2$ in words. Nonetheless, the calculation is involved in practice and there is a merit to have these worked out in detail.

Weaknesses

Being an extension of the analysis of the reference $[1]$, the manuscript shares some of the technical subtleties, appearing in the reference $[1]$. See the report below.

Report

  1. Why don't we have complementary series (or possible other special representations, depending on $d$) representation appearing in Eq 2.2 ?

  2. In the paragraph following the paragraph containing Eq $2.2$, the authors mention of unitary representations of $SO(d+1,1)$. The relevant sentence seems to suggest that everything $\Delta>d/2$ is unitary representation of $SO(d+1,1)$ while the ones with $\Delta<d/2$ are not. This is not true. $SO(d+1,1)$ has principal and complementary series as unitary representations. For example, in $d=2$, they exhaust all the unitary representations of $SO(3,1)$. The complementary series is labeled by $\Delta\in(0,d)$ and there is a unitary isomorphism between $\Delta$ and $(d-\Delta)$. A clarification in this regard would help the readers.

  3. In Eq $2.2$, the notion of descendants is very confusing in the context of Principal series. The Principal series representation is not of the lowest or highest weight kind, this is a standard result in the representation theory of $SO(d+1,1)$. So it would be nice to clarify what is meant by descendant in Eq $2.2$. On the same note, it's not clear how $V(x,z)$ is defined. Usually coherent states $O(z)$ in CFT are well defined because $O(0)$ is a lowest weight state. Indeed such lowest weight representations exist in usual Lorentzian unitary CFT. But as far as the unitary Principal representation of $SO(d+1,1)$ is concerned, no such normalizable state exists. One way out to be to say there is no state operator correspondence in this case, however, then I would be worried about OPE convergence. In short, it would be nice to clarify the construction of $V(x,z)$. In context of AdS$_3$-CFT$_2$, people study $H_3^+$ model, construction of $V(x,z)$ is worked out in the classic paper of Teschner [see https://arxiv.org/abs/hep-th/9712256] . Is there something analogous in every dimension?

  4. The analytic continuation of Eq $2.2$ is very mysterious. It is not clear to me at all what is meant and how it is done for general dimension. Again in the paper of Teschner, an assumption on continuity was important, which is later used and emphasized in the classic paper by Hirosi and Juan [See https://arxiv.org/abs/hep-th/0111180] . Furthermore, in figure $1$ (even though it is borrowed from Ref $[1]$), there can be poles of the integrand in between two vertical lines away from the real line. This possibility is not addressed in Ref $[1]$ as well. I don't see why such possibility is excluded. Such a pole would immediately call into question Lorentzian unitarity of the boundary CFT. A comment in this regard would be useful.

  5. Assuming the descendants [see the point above] make sense, I think that authors assumed uniform convergence of OPE while dropping the contribution from the descendants under the $z$ integral. Even if the contribution from the descendants are suppressed before doing the integral over $z$, it does not guarantee that they are still suppressed after doing the integral over $z$, especially when the integral over $\Delta$ involves shifting the contour. In fact, away from the Principal series line, I would really worry about the OPE convergence.

  6. There is also a swap of $\Delta$ and $z$ integral. I am not sure why it is justified apriori. Furthemore, the saddle point analysis is done sequentially. In multivariable saddle point analysis, it is important to do an analysis involving two variables simultaneously and then compute the Hessian matrix. Importantly, the Hessian matrix can have off-diagonal elements. Doing the saddle point analysis sequentially amounts to computing the two diagonal elements of this matrix.

  7. Later on, when the authors talk about boundary CFT operators with spin, it's not clear whether they talk about $SO(d+1,1)$ or universal cover of Lorentzian conformal group i.e $\widetilde{SO(d,2)}$. In section 5, there is statement of the form \textit{The conformal dimension of the conserved vector operators is $d-1$}. This led me to believe that the authors are talking about Lorentzian conformal group. Is it correct?

Requested changes

It would be nice to make some comments regarding the points raised in the report.

---

## Editorial Decision

awaiting_resubmission